# RNA m⁶A methylation regulates sorafenib resistance in liver cancer through FOXO3-mediated autophagy

Ziyou Lin[1,2], Yi Niu[2], Arabella Wan[1], Dongshi Chen[3], Heng Liang[2], Xijun Chen[4], Lei Sun[2], Siyue Zhan[2], Liutao Chen[5], Chao Cheng[6], Xiaolei Zhang[2], Xianzhang Bu[2], Weiling He[1,2,7,*] & Guohui Wan[1,2,**]

## Abstract

N6-methyladenosine (m⁶A) is an abundant nucleotide modification in mRNA, known to regulate mRNA stability, splicing, and translation, but it is unclear whether it is also has a physiological role in the intratumoral microenvironment and cancer drug resistance. Here, we find that METTL3, a primary m⁶A methyltransferase, is significantly down-regulated in human sorafenib-resistant hepatocellular carcinoma (HCC). Depletion of METTL3 under hypoxia promotes sorafenib resistance and expression of angiogenesis genes in cultured HCC cells and activates autophagy-associated pathways. Mechanistically, we have identified FOXO3 as a key downstream target of METTL3, with m⁶A modification of the FOXO3 mRNA 3′-untranslated region increasing its stability through a YTHDF1-dependent mechanism. Analysis of clinical samples furthermore showed that METTL3 and FOXO3 levels are tightly correlated in HCC patients. In mouse xenograft models, METTL3 depletion significantly enhances sorafenib resistance of HCC by abolishing the identified METTL3-mediated FOXO3 mRNA stabilization, and overexpression of FOXO3 restores m⁶A-dependent sorafenib sensitivity. Collectively, our work reveals a critical function for METTL3-mediated m⁶A modification in the hypoxic tumor microenvironment and identifies FOXO3 as an important target of m⁶A modification in the resistance of HCC to sorafenib therapy.

**Keywords** autophagy; FOXO3; hypoxia; METTL3; N6-methyladenosine
**Subject Categories** Cancer; RNA Biology
**The EMBO Journal (2020) 39: e103181**

## Introduction

Hepatocellular carcinoma (HCC) is a primary liver malignancy in patients with chronic liver disease and cirrhosis (Villanueva, 2019), and compared with most solid tumors its incidence is increasing worldwide year by year due to hepatitis viruses (HBV and HCV) infection and alcohol use (Siegel *et al*, 2019). In the early stages, curative treatments such as tumor resection, ablation, and liver transplantation can be used to treat HCC (Yang *et al*, 2019). Patients with advanced HCC are administrated with topical treatment and systemic chemotherapy as they are no longer eligible for curative therapies. Sorafenib, the only FDA-approved first-line treatment for advanced HCC, is a multi-target kinase inhibitor for Raf kinases, vascular endothelial growth factor, and platelet-derived growth factor receptors, and improves survival with a median OS rate of 6.5 months compared to the placebo group (Cheng *et al*, 2009). However, patients with advanced HCC predominantly develop resistance to sorafenib treatment and their survival benefit is limited to 3–5 months with severe side effects (Llovet *et al*, 2008). A hypoxic microenvironment is a common feature of human advanced solid tumors linked to resistance. The median O₂ partial pressure in HCC is 6 mm Hg compared with 30 mm Hg in normal liver (Vaupel *et al*, 2007), and the homeostatic response to the intratumoral hypoxic environment is mediated by hypoxia-inducible factor (HIF-1), consisting of HIF-1α and HIF-1β subunits (Koh & Powis, 2012). Examining the underlying mechanisms of acquired resistance toward sorafenib in the intratumoral hypoxic microenvironment may provide a new approach to develop individualized therapeutic strategies for coping with sorafenib resistance and to investigate potential combination therapies for the advanced HCC.

Recently, the role of N6-methyladenosine (m⁶A) regulation in various biological processes has been an emerging focus of investigation. m⁶A is a predominantly internal modification of RNA in mammalian cells, with the features of being dynamic and reversible

1    Department of Gastrointestinal Surgery, The First Affiliated Hospital, Sun Yat-Sen University, Guangzhou, China
2    National-Local Joint Engineering Laboratory of Druggability and New Drug Evaluation, National Engineering Research Center for New Drug and Druggability (cultivation), Guangdong Province Key Laboratory of Chiral Molecule and Drug Discovery, School of Pharmaceutical Sciences, Sun Yat-Sen University, Guangzhou, China
3    Division of Pulmonary, Allergy and Critical Care Medicine, Department of Medicine, University of Pittsburgh, Pittsburgh, PA, USA
4    Department of Abdominal Surgery, Integrated Hospital of Traditional Chinese Medicine, Southern Medical University, Guangzhou, China
5    School of Life Science, Sun Yat-Sen University, Guangzhou, China
6    Department of Thoracic Surgery, The First Affiliated Hospital, Sun Yat-Sen University, Guangzhou, China
7    Center for Precision Medicine, Sun Yat-Sen University, Guangzhou, China
    *Corresponding author. Tel: +86 20 87755766; E-mail: hewling@mail.sysu.edu.cn
    **Corresponding author. Tel: +86 20 39943495; E-mail: wanguoh@mail.sysu.edu.cn

(Niu *et al*, 2018). The functional components of the RNA methyl-transferase complex include METTL3 (methyltransferase-like 3), METTL14 (methyltransferase-like 14), and WTAP (Wilms tumor 1-associated protein), while the RNA demethylases include FTO (fat mass and obesity-associated protein) and ALKBH5 (a-ketoglutarate-dependent dioxygenase alkB homolog 5). $m^6A$ binding proteins with a YT521-B homology (YTH) domain, including YTHDC1, YTHDC2, YTHDF1, YTHDF2, and YTHDF3, recognize $m^6A$ in a methylation-dependent manner (Zaccara *et al*, 2019). Furthermore, the $m^6A$ methyltransferase function of METTL3 strictly requires METTL14 as a co-activator. METTL3 and METTL14 form a $m^6A$ holoenzyme complex, where METTL3 functions as the catalytic subunit, while METTL14 binds to RNA substrates, stabilizes the structure of the complex, and activates METTL3 via allostery (Sledz & Jinek, 2016; Wang *et al*, 2016a,b). Emerging evidence has shown that $m^6A$ modification plays important roles in various cellular processes including RNA stability (Schwartz *et al*, 2014; Wang *et al*, 2014), translation (Wang *et al*, 2015; Zhou *et al*, 2015), structure (Liu *et al*, 2015), localization (Fustin *et al*, 2013), alternative polyadenylation, and splicing (Haussmann *et al*, 2016; Lence *et al*, 2016). Dysregulation of $m^6A$ methylation has been frequently reported in various human cancers (Cui *et al*, 2017; Li *et al*, 2017; Liu *et al*, 2018; Niu *et al*, 2019). For example, we previously showed that FTO induced breast cancer progression through demethylation of BNIP3 in a YTHDF2-independent manner (Niu *et al*, 2019). Targeting key regulators of the $m^6A$ modification has been discussed as a potential therapeutic approach for cancer treatment (Niu *et al*, 2018). However, the role of the $m^6A$ modification in drug resistance of HCC has not been fully described. Thus, examining the $m^6A$ regulation in sorafenib-resistant HCC under intratumoral hypoxic microenvironment can provide more comprehensive insights into the molecular mechanisms of the occurrence of sorafenib resistance in HCC.

Herein, we discovered that the $m^6A$ modification was decreased in sorafenib-resistant HCC and in sorafenib-resistant liver tumor cells. Depletion of METTL3, the primary $m^6A$ methyltransferase, enhanced HCC resistance to sorafenib through activating angiogenesis and autophagy pathways. Using MeRIP-Seq analysis and functional studies, we identified FOXO3 as a critical downstream target of METTL3-mediated $m^6A$ modification. Our results showed that METTL3 promoted FOXO3 mRNA methylation in the 3′UTR region and enhanced its mRNA stability in an YTHDF1-dependent manner. Overexpression of FOXO3 rescued the sorafenib-resistant phenotype induced by METTL3 depletion. We further validated the effects of the METTL3/FOXO3 axis on sorafenib resistance through both *in vitro* and *in vivo* experiments. Taken together, our study reveals a connection of $m^6A$ modification, sorafenib resistance, and autophagy under hypoxia, and provides insights into the multiple molecular mechanisms of sorafenib resistance in HCC, as well as expanding the understanding of therapy resistance.

# Results

### Down-regulation of METTL3 in sorafenib-resistant HCCs

To investigate the molecular mechanism of sorafenib resistance in HCC, we obtained liver tumors with acquired sorafenib resistance ($n = 3$) from patients with long-term sorafenib treatment and performed transcriptome sequencing to examine the differentially expressed genes by comparison to sorafenib-sensitive liver tumors ($n = 3$). We identified 819 genes with significant up-regulation and 775 genes with significant down-regulation in the sorafenib-resistant liver tumors compared to the sorafenib-sensitive liver tumors (Appendix Fig S1A and Table EV2). Gene enrichment analysis with KOBAS highlighted dysregulation of several signaling pathways involved in drug resistance, including cellular response to chemical stimulus, response to organic substance, and response to nitrogen compound (Fig 1A). A novel group of transferase was identified to play roles in sorafenib resistance in liver cancer. 13 genes overlapped in these four signaling pathways (Fig 1B). We identified that METTL3, the primary component of $m^6A$ methyltransferase, was significantly down-regulated in sorafenib-resistant HCC (Fig 1C). We further validated down-regulation of METTL3 in sorafenib-resistant liver tumors by RT–PCR (Fig 1D). In a clinical outcome analysis, we found that down-regulation of METTL3 was associated with lower survival rates in HCC patients (Appendix Fig S1B). By analyzing the pathological stage plot in liver cancer in the TCGA database, we found that METTL3 expression level was significantly down-regulated in advanced stages of HCCs (Fig 1E). These results indicate that down-regulation of METTL3 may be implicated in sorafenib resistance in HCC.

To confirm the role of METTL3 in sorafenib resistance, we systematically analyzed the GSE62813 database for the change of METTL3 expression in HepG-2 cells and sorafenib-resistant HepG-2 cells, and found that METTL3 was significantly down-regulated in sorafenib-resistant HCC cells (Fig 1F). To validate the role of

**Figure 1. METTL3 was down-regulated in the sorafenib-resistant HCCs.**

A Gene enrichment analysis on differentially expressed genes between sorafenib-sensitive and sorafenib-resistant human liver tumors.
B Venn diagrams show overlapped genes in signaling pathways response to sorafenib.
C Heatmap of overlapped expressed genes in response to sorafenib resistance.
D The mRNA expression level of METTL3 in sorafenib-sensitive ($n = 3$) and sorafenib-resistant ($n = 3$) human liver tumors.
E Expression of METTL3 was significantly down-regulated in advanced stage of liver cancer from TCGA.
F METTL3 RNA levels between naïve HepG-2 cells ($n = 3$) and sorafenib-resistant HepG-2 cells ($n = 10$).
G The IC50 of sorafenib-resistant HepG-2 cells treated with sorafenib under hypoxic condition (1% $O_2$).
H The protein level of METTL3 in sorafenib-resistant HepG-2 cells treated with sorafenib under hypoxic condition (1% $O_2$).
I The global RNA $m^6A$ level in naïve HepG-2 and sorafenib-resistant HepG-2 determined by dot blotting assay under hypoxic condition (1% $O_2$).

Data information: In all relevant panels, \***$P < 0.001$; \****$P < 0.0001$; two-tailed *t*-test. Data are presented as mean ± SD and are representative of three independent experiments.
Source data are available online for this figure.

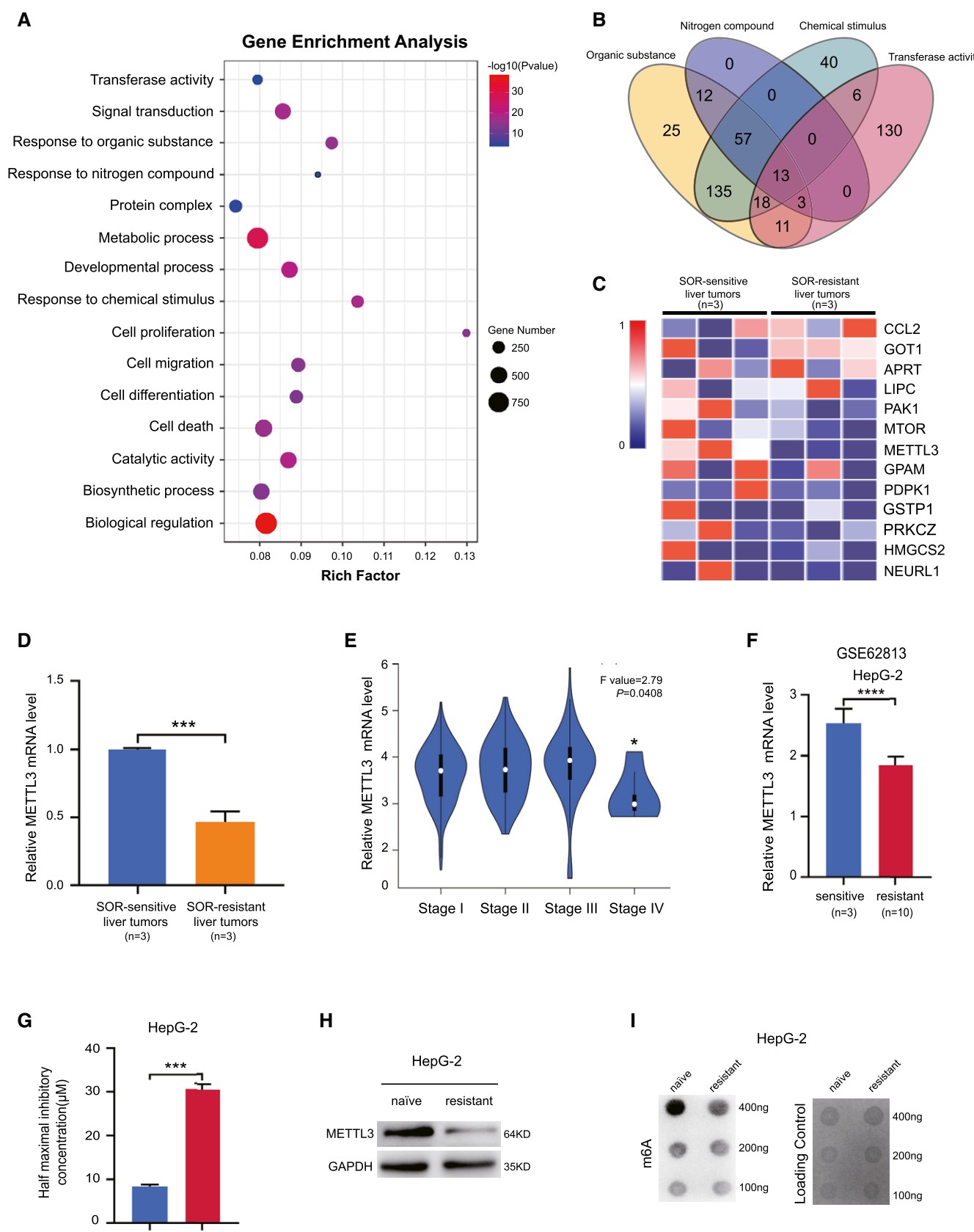

**Figure 1.**

METTL3 in acquired resistance, we generated a sorafenib-resistant HepG2 cell line *in vitro* by exposing the cells with sorafenib at 5% of IC concentration for 3 days and gradually increased the concentration by 5% of IC until reaching the IC50 concentration. We confirmed the acquired resistance of these HepG2 cells toward sorafenib by comparing to the naïve HepG-2 cells (Fig 1G). Importantly, we found that METTL3 was markedly reduced in sorafenib-resistant HepG-2 cells with consistently decreased global RNA m$^6$A level (Fig 1H and I), indicating a potential role of METTL3 in mediating resistance toward sorafenib in HCC cells.

Moreover, we noticed that hypoxic condition extensively existed in HCC, as we observed an elevated expression level of HIF-1α in liver tumors when comparing to non-tumor liver tissues (Appendix Fig S1C and D). We established a HCC subcutaneous tumor model in nude mice with SMMC-7721 cells and found that 80% of tumors exhibited a higher level of hypoxia (Appendix Fig S1E and F). We also performed an orthotopic liver tumor xenograft with Bel-7402 cells to validate the hypoxic condition in HCC. Higher levels of HIF-1α were detected in tumor tissue compared to non-tumor tissue (Appendix Fig S1G and G1), indicating oxygen deprivation was a common feature of HCC. To examine the role of METTL3 in HCC, our following functional experiments *in vitro* were conducted at low oxygen levels (1% $O_2$) to mimic the intratumoral microenvironment in HCC.

## Knockdown of METTL3 enhanced sorafenib resistance in HCC

To evaluate whether the global m$^6$A level is related to tumorigenesis in liver cancer, we generated a series of various expression levels of METTL3 in the normal liver cell line WRL68 (Appendix Fig S2A and B). The normal liver cells formed an enlarged colony after silencing METTL3, but had no meaningful change after overexpressing METTL3 (Fig 2A and B). Neither METTL3 knockdown nor METTL3 overexpression in WRL68 cells induced sensitivity toward sorafenib treatment, indicating that sorafenib was not toxic to normal liver cells regardless of various levels of METTL3 expression (Appendix Fig S2C). However, knockdown of METTL3 in WRL68 cells moderately promoted cell growth, while overexpression of

METTL3 in WRL68 cells had no effect in a cell proliferation assay and cell number counting (Appendix Fig S2D and E). To further explore the role of the m$^6$A modification in HCC toward sorafenib resistance, we constructed six stable METTL3-depleted HCC cell lines (SMMC-7721 #sh1 #sh2, Bel-7402 #sh1 #sh2, and HepG-2 #sh1 #sh2) and confirmed the knockdown effects in both protein and RNA expression levels (Appendix Fig S2F and G). We showed the decrease of global RNA m$^6$A level in these six stable METTL3-depleted cell lines by dot blot assay (Appendix Fig S2H), and the increased sensitivity (IC50) of METTL3-knockdown SMMC-7721, Bel-7402, and HepG-2 cells to treatment with sorafenib (Fig 2C and D, Appendix Fig S2I). We further confirmed the resistant phenotypes of METTL3-knockdown HCC cells toward sorafenib by cell survival assays (Appendix Fig S2J–L) under hypoxia. We noticed a moderately increased IC50 for sorafenib treatment of METTL3-knockdown HCCs under normoxic condition (21% $O_2$) (Appendix Fig S2M–O), excluding that the potential molecular mechanism of METTL3-mediated sorafenib resistance in HCC was mainly caused by hypoxia.

Moreover, to determine the role of m$^6$A in sorafenib resistance in HCC, we performed rescue experiments in HCC cells with stable METTL3 depletion and sorafenib-resistant HepG-2 cells using wild-type METTL3 and a catalytic mutant METTL3 (resistant to METTL3 shRNA) (Appendix Fig S2P and Q) (Fig 2E and F). Our results showed that rescue of shRNA-resistant wild-type METTL3, but not catalytic mutant METTL3, sensitizes METTL3-knockdown HCC cells (Fig 2G and H) and sorafenib-resistant HepG-2 cells (Fig 2I) to sorafenib treatment. Consistently, the results of CCK8 cell growth assay in HCCs with METTL3 knockdown (Appendix Fig S2R and S) or sorafenib resistance (Appendix Fig S2T) showed a similar rescuing tendency, as well as a cell survival assay in sorafenib-resistant HepG-2 cells (Fig 2J), indicating the important role of the m$^6$A modification mediated by METTL3 in sorafenib resistance in HCC.

To further determine how METTL3 controls sorafenib resistance in HCC, we determined the influence of METTL3 on the tube-forming ability of human umbilical vein endothelial cells (HUVECs). Our results revealed that the tumor-conditioned medium (TCM)

---

**Figure 2. Knockdown of METTL3 enhanced sorafenib resistance in HCC.**

A, B   Clonogenic survival of METTL3 knockdown (A) and METTL3 overexpression (B) in normal liver cell line WRL68 cells for 7 days in normoxic condition (21% $O_2$) and quantification of clusters in A-1 and B-1, respectively.

C, D   The IC50 of METTL3-knockdown SMMC-7721 cells (C) and Bel-7402 cells (D) after treated with sorafenib for 24 h under hypoxic condition (1% $O_2$).

E   Overexpression of wild-type METTL3 or catalytic mutant METTL3 in sorafenib-resistant HepG-2 cells.

F   The global RNA m$^6$A level in sorafenib-resistant HepG-2 cells with wild-type METTL3 overexpression or catalytic mutant METTL3 overexpression by dot blotting assay.

G–I   Rescue of shRNA-resistant wild-type METTL3 but not catalytic mutant METTL3 sensitized METTL3-knockdown SMMC-7721 cells (G), Bel-7402 cells (H), and sorafenib-resistant HepG-2 cells (I) to sorafenib treatment. The IC50 of the cells was measured after treated with sorafenib for 24 h under hypoxic condition (1% $O_2$).

J   Cell survival assay of sorafenib-resistant HepG-2 cells with wild-type METTL3 overexpression or catalytic mutant METTL3 overexpression after treated with sorafenib for 24 h under hypoxic condition (1% $O_2$). Scale bar, 1 mm.

K   Capillary-like structures in HUVECs treated with the tumor-conditioned medium (TCM) from control HCCs and METTL3-knockdown HCCs cultured under hypoxic conditions for 48 h. Quantification of the number of tubes with ImageJ in K-1.

L, M   The relative mRNA expression levels of angiogenesis genes were detected in SMMC-7721 cells (L) and Bel-7402 cells (M) with METTL3 knockdown by RT–PCR assay.

N   The protein levels of VEGF-A in METTL3-knockdown HCCs.

O   The protein levels of PDGF-B in METTL3-knockdown HCCs.

Data information: In all relevant panels, *$P < 0.05$; **$P < 0.01$; ***$P < 0.001$; ****$P < 0.0001$; two-tailed *t*-test. Data are presented as mean ± SD and are representative of three independent experiments.
Source data are available online for this figure.

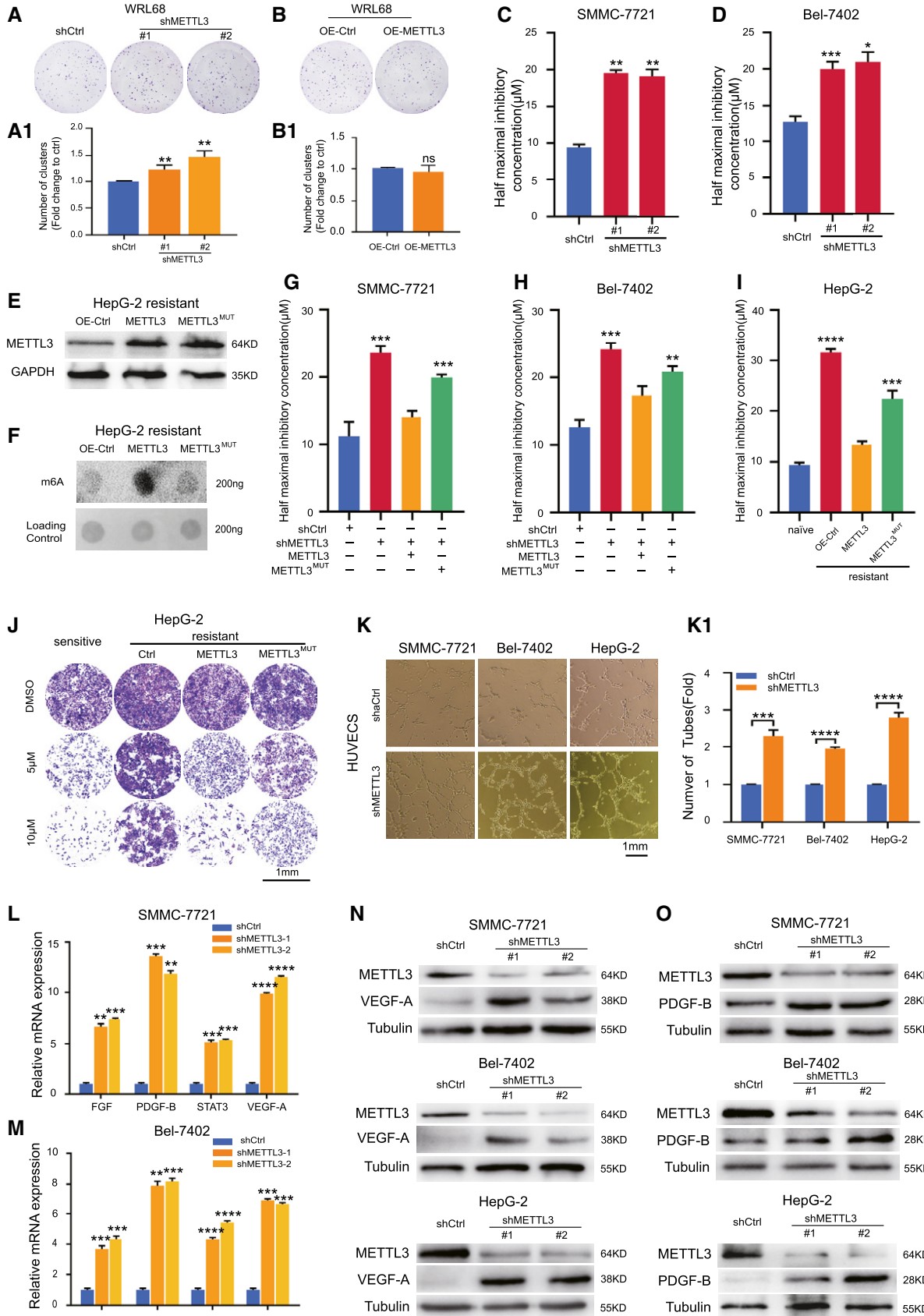

**Figure 2.**

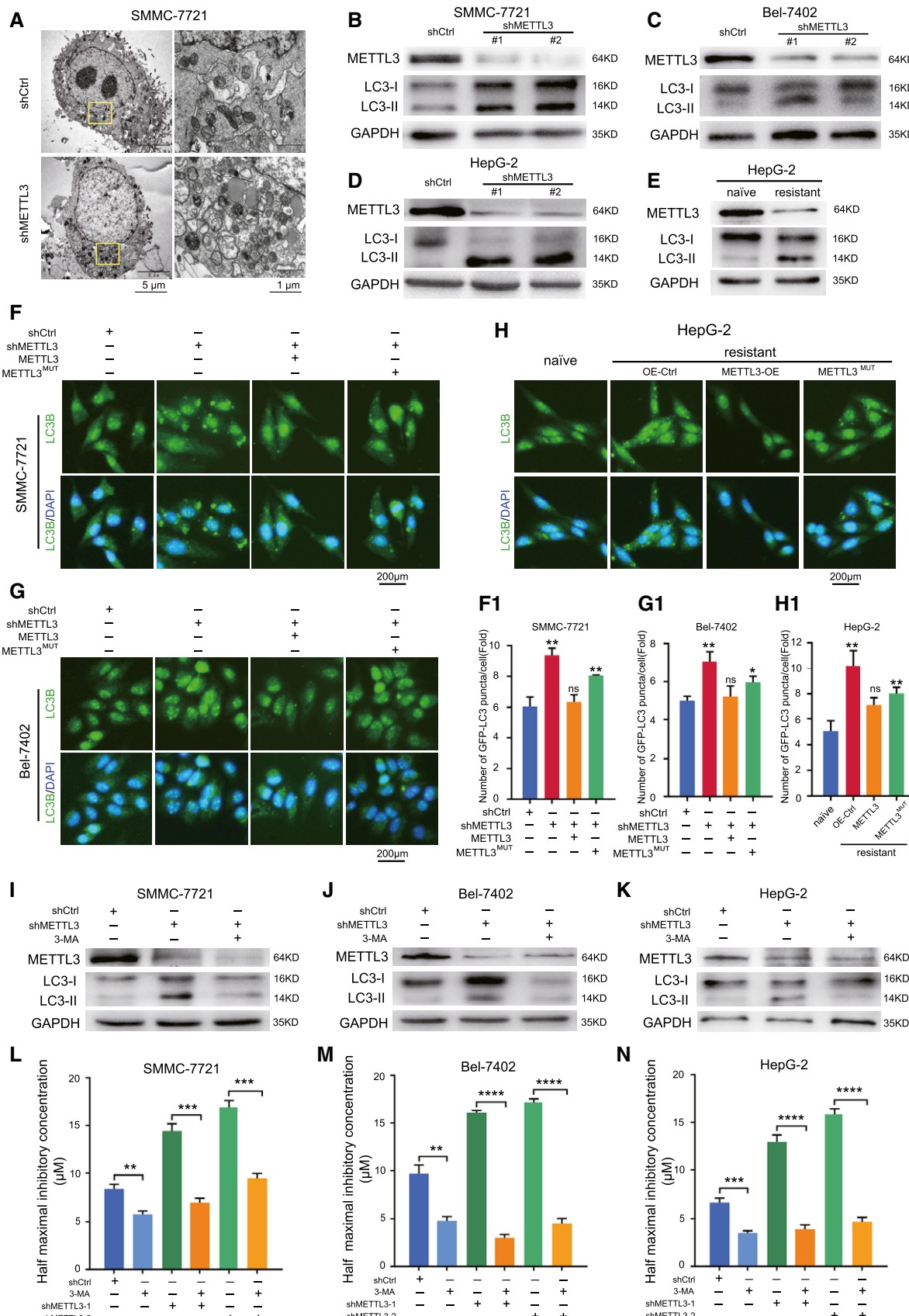

**Figure 3.**

**Figure 3.   METTL3-dependent sorafenib resistance in HCC was mediated by promoting autophagy in HCC.**

A       Electron micrographs of METTL3-knockdown SMMC-7721 cells under hypoxia for 48 h.
B–D    Protein levels of LC3 I/II in METTL3-knockdown SMMC-7721 (B), Bel-7402 (C), and HepG-2 (D) cells under hypoxic condition for 48 h.
E       Protein level of LC3 I/II in HepG-2 and sorafenib-resistant HepG-2 cells under hypoxic condition for 48 h.
F–H    Overexpression of shRNA-resistant wild-type METTL3 but not catalytic mutant METTL3 rescue autophagy phenotype in METTL3-knockdown SMMC-7721 cells (F),
        Bel-7402 cells (G), and sorafenib-resistant HepG2 cells (H) by representative immunostaining images of LC3 under hypoxic condition for 48 h. Scale bar, 200 μm.
        Quantification of the numbers of GFP-LC3 puncta/cell by Imaris in F-1, G-1, and H-1, respectively.
I–K    Protein levels of LC3 I/II in METTL3-knockdown SMMC-7721 cells (I), Bel-7402 cells (J), and HepG-2 cells (K) treated with or without 3-MA under hypoxic condition
        for 24 h.
L–N    The IC50 of sorafenib in SMMC-7721 cells (L), Bel-7402 cells (M), and HepG-2 cells (N) treated with or without 3-MA for 24 h under hypoxic condition (1% $O_2$).

Data information: In all relevant panels, *$P < 0.05$; **$P < 0.01$; ***$P < 0.001$; ****$P < 0.0001$; two-tailed $t$-test. Data are presented as mean ± SD and are representative
of 3 independent experiments.
Source data are available online for this figure.

from HCC cells in the METTL3-knockdown groups showed increased capacity to stimulate HUVEC tube formation compared to the control groups (Fig 2K and K-1). Moreover, knockdown of METTL3 up-regulated RNA expression of some biomarkers of angiogenesis such as FGF, PDGF-B, STAT3, and VEGF-A in HCC cells under hypoxia (Fig 2L and M, Appendix Fig S2U). We further confirmed the induced protein levels of VEGF-A and PDGF-B by Western blot (Fig 2N and O). Taken together, our results demonstrate that METTL3 deletion enhanced sorafenib resistance in HCC.

**Autophagy aberration mediated METTL3-dependent sorafenib resistance in HCC**

Previous reports have shown that autophagy was associated with chemo-resistance in human cancers (Levine & Kroemer, 2008; Doherty & Baehrecke, 2018). To determine whether m⁶A regulation played a role in autophagy to mediate sorafenib resistance in HCC, transmission electron microscopy (TEM) was used to observe the morphological changes in SMMC-7721 cells with or without METTL3 depletion under hypoxia. Our results showed that knockdown of METTL3 increased the number of autophagosomes (Fig 3A), suggesting ongoing autophagy. To further confirm this phenomenon, we measured the levels of lipidation of LC3, a biomarker in autophagy. Our results showed that knockdown of METTL3 significantly increased LC3-II accumulation in HCC cell lines (Fig 3B–D). LC3-II accumulation was also enhanced in sorafenib-resistant HepG-2 cells, which had relatively lower expression of METTL3 when comparing to the parental cells (Fig 3E). Interestingly, we analyzed the subcellular redistribution of GFP-LC3 by fluorescent immunostaining under normoxic conditions and found a moderate increase in their number in METTL3-depleted groups compared to the control groups (Appendix Fig S3A and B). Notably, we detected the increased subcellular redistribution of GFP-LC3 in METTL3-knockdown HCC cells and sorafenib-resistant HepG-2 cells by fluorescent immunostaining. Rescue of wild-type METTL3, but not catalytic mutant METTL3, reduced the accumulation of GFP-LC3 in these cells (Fig 3F–H and Appendix Fig S3C and D). To examine the role of autophagy in sorafenib resistance caused by METTL3 depletion, we measured the IC50 of sorafenib in METTL3-knockdown HCC cells combined with 3-MA, an inhibitor of autophagy. Treatment with 3-MA reduced LC3-II accumulation mediated by knockdown of METTL3 (Fig 3I–K and Appendix Fig S3E) and caused a dramatic reduction of the IC50 of sorafenib in groups with

3-MA treatment (Fig 3L–N). These results suggested that METTL3 depletion augmented the autophagic flux toward sorafenib-resistance.

**METTL3 regulates autophagy in HCC through mediating FOXO3 signaling**

To explore the mechanism by which METTL3 depletion activates autophagy in HCC, we performed an autophagy real-time quantitative PCR array to profile gene expression in METTL3-knockdown HCC cells and observed good consistent results between the two independent shRNAs of METTL3 in three HCC cell lines (Appendix Fig S4A). Gene set enrichment analysis (GSEA) revealed enrichment of pathway in cellular response to stress genes, which was defined as unfavorable environmental conditions such as metabolism, osmotic stress, and oxidative stress like hypoxia. Twenty-three genes with significant changes were enriched in GSEA, and seven genes showed consistent tendency across the three HCC cell line samples, including PIK3C3, RELA, FOXO3, MTOR, ATG5, MAP3K7, and MAPK1 (Fig 4A and Appendix Fig S4A). KEGG analysis of the hypoxia-induced autophagy pathway identified FOXO3 as a universal transcription factor with a significant change in autophagic signaling pathway when METTL3 was knocked down (Fig 4B), and STRING analysis showed that FOXO3 was involved in regulating key biomarkers in autophagy (Appendix Fig S4B).

To determine how METTL3 regulated FOXO3 in HCC, we first examined the expression level of FOXO3 upon METTL3 knockdown. We confirmed that FOXO3 was dramatically down-regulated in both protein and RNA levels in stable METTL3-knockdown cells (Fig 4C and Appendix Fig S4C). Rescue with wild-type METTL3, but not the catalytic mutant METTL3, restored the expression level of FOXO3 in METTL3-knockdown HCC cells (Appendix Fig S4D and E) and sorafenib-resistant HepG-2 cells (Appendix Fig S4F), suggesting a role of m⁶A-modification mediated by METTL3 in regulating FOXO3 expression level. After treatment with actinomycin D and cycloheximide, respectively, the half-life of FOXO3 mRNA and the protein level was remarkably decreased in METTL3-knockdown cells compared to control cells, indicating that abolishment of FOXO3 m⁶A methylation could accelerate its degradation (Fig 4D and E). Interestingly, we detected a slight increase of polysome processing in METTL3-knockdown cells under hypoxia (Appendix Fig S4G). By separating the RNA fractions from non-translating fraction (< 40S), translation initiation fraction (40S, 60S, 80S, and < 80S) to translation active polysomes (> 80S), we found that knockdown of

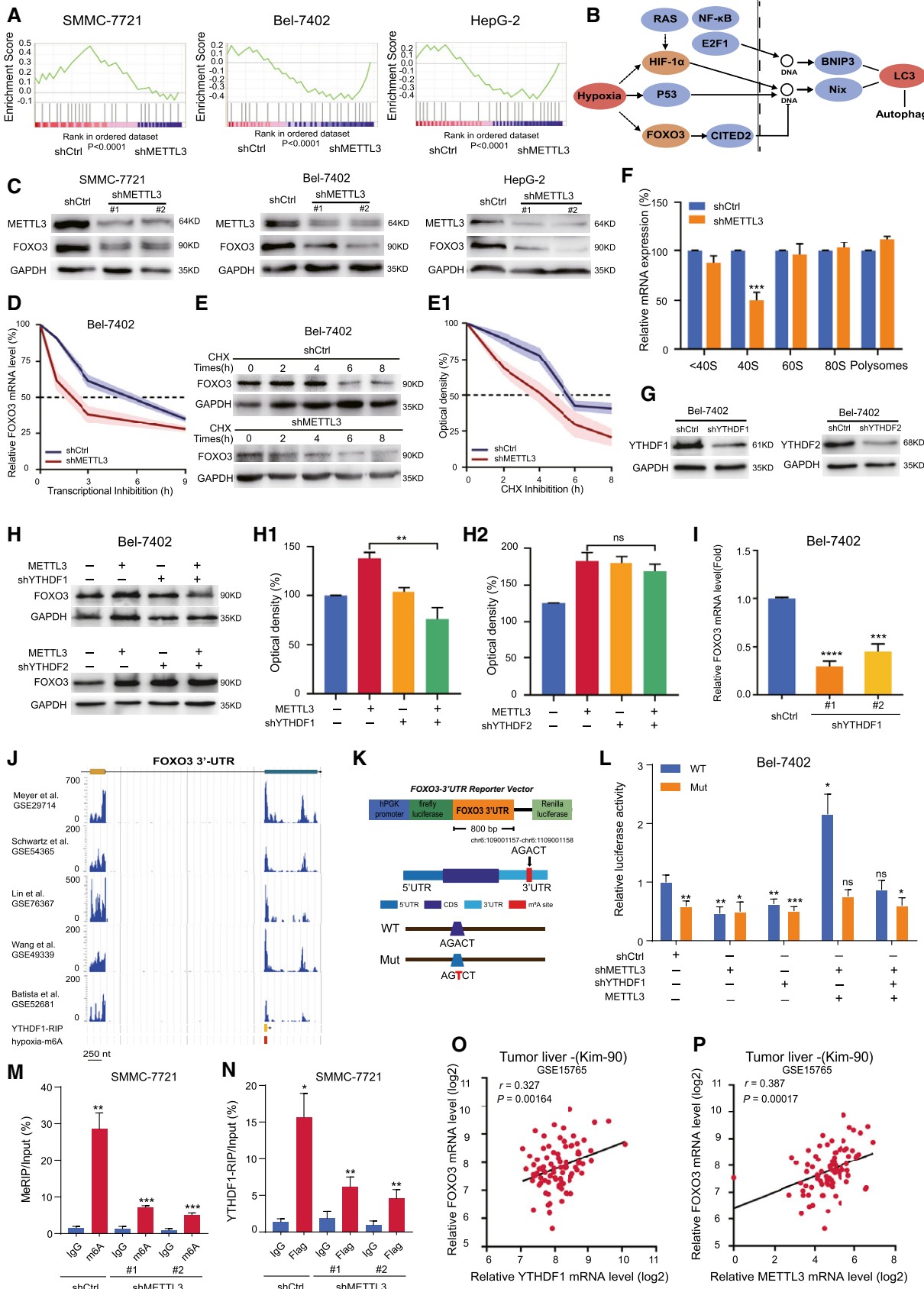

Figure 4.

**Figure 4.  RNA m⁶A-Seq identified FOXO3 as a downstream target of METTL3-mediated m⁶A modification under hypoxia.**

A   Gene set enrichment analysis (GSEA) output images of cellular response to stress pathways displaying a correlation of differentially regulated genes in METTL3-knockdown HepG2, SMMC-7721, and Bel-7402 cells with the C5, biological process symbol set ($P < 0.001$).
B   The KEGG analysis shows that FOXO3 involved in hypoxia and autophagy.
C   Protein level of FOXO3 in METTL3-knockdown HCCs under hypoxia for 48 h.
D   Half-life of FOXO3 mRNA in Bel-7402 cells and correspondent METTL3-knockdown Bel-7402 cells treated with actinomycin D.
E   Half-life of FOXO3 protein in Bel-7402 cells and correspondent METTL3-knockdown Bel-7402 cells treated with cycloheximide. Quantification of the protein optical density by ImageJ in E-1.
F   Analysis of FOXO3 mRNA in non-ribosome portion (<40S), 40S, 60S, 80S, and polysome for the METTL3-knockdown Bel-7402 cells compared to control cells under hypoxia for 48 h.
G   Knockdown of YTHDF1 and YTHDF2 in Bel-7402 cells by lentiviral shRNAs.
H   Knockdown of YTHDF1 but not YTHDF2 abolished the increase of FOXO3 protein level in METTL3-overexpressing Bel-7402 cells. Quantification of the protein optical density by ImageJ in H-1 and H-2.
I   RNA level of FOXO3 in YTHDF1-knockdown Bel-7402 cells by RT–PCR.
J   The m⁶A-IP sequencing under hypoxia proved the m⁶A modification participated in regulation of FOXO3. The YTHDF1 binding site locates at the 3′UTR of FOXO3.
K   800 bp of FOXO3 3′UTR containing the m⁶A modification site was cloned into luciferase reporter vector. Mutation of m⁶A consensus sequence was generated by replacing adenosine with thymine.
L   Relative luciferase activity of the wild-type and mutant form of FOXO3 3′UTR reporter vectors in Bel-7402 cells with various levels of METTL3 and YTHDF1.
M   Knockdown of METTL3 reduced the m⁶A methylation in FOXO3 mRNA by the m⁶A MeRIP analysis.
N   Knockdown of METTL3 reduced the m⁶A methylation in FOXO3 mRNA by the YTHDF1-RIP analysis.
O   Correlation analysis of the RNA expression levels of METTL3 and FOXO3 in liver tumor cohort (Kim-90) by Pearson.
P   Correlation analysis of the RNA expression levels of YTHDF1 and FOXO3 in liver tumor cohort (Kim-90) by Pearson.

Data information: In all relevant panels, *$P < 0.05$; **$P < 0.01$; ***$P < 0.001$; ****$P < 0.0001$; two-tailed $t$-test. Data are presented as mean $\pm$ SD and are representative of three independent experiments.
Source data are available online for this figure.

METTL3 significantly decreased translation efficiency of FOXO3 at 40S fraction (Fig 4F). Next, to determine the potential m⁶A reader that regulates the expression level of FOXO3, we knocked down YTHDF1 and YTHDF2 in Bel-7402 cells (Fig 4G). Knockdown of YTHDF1, but not YTHDF2, abolished the increased protein level of FOXO3 in METTL3-overexpressing Bel-7402 cells (Fig 4H). In turn, depletion of YTHDF1 significantly reduced the RNA expression level of FOXO3 (Fig 4I), indicating that YTHDF1 may be involved in regulation of FOXO3 expression through m⁶A modification.

Next, we performed the m⁶A-Seq to map the m⁶A sites in SMMC-7721 cells under hypoxia and found that the m⁶A site of FOXO3 is located at the 3′UTR region, which overlaps with a YTHDF1-binding site previously identified by YTHDF1-RIP assay (Wang *et al*, 2015). Consistent with our results, previous studies had identified FOXO3 mRNA as a direct target of the m⁶A modification (Meyer *et al*, 2012; Batista *et al*, 2014; Schwartz *et al*, 2014; Wang *et al*, 2014; Lin *et al*, 2016). Metagene analysis of FOXO3 at a spring-loaded base modification further confirmed the m⁶A site at the 3′UTR region (Appendix Fig S4H). To prove the effect of m⁶A regulation on FOXO3 expression mediated by YTHDF1, we cloned the FOXO3 3′UTR into a dual-luciferase reporter construct and generated the mutant form of FOXO3 3′UTR by replacing A with T in the core motif (RRACH) of the potential m⁶A site (Fig 4K). Relative normalized luciferase activities of the wild-type and the mutant form of FOXO3 3′UTR were measured in Bel-7402 cells with various levels of METTL3. A reduction of luciferase activity was detected for the wild-type 3′UTR of FOXO3 in METTL3-knockdown and YTHDF1-knockdown cells. In contrast, for the mutant form of 3′UTR of FOXO3, depletion of METTL3 or YTHDF1 does not have any effect (Fig 4L). In METTL3-overexpressing cells, depletion of YTHDF1 abolished the induction of luciferase activity for the wild-type form of FOXO3 (Fig 4L). Further, to validate FOXO3 as a bona fide target for METTL3-mediated m⁶A modification under hypoxia, we performed an m⁶A-RNA immunoprecipitation assay (MeRIP) and

YTHDF1-RIP assay and analyzed these by RT–PCR. Knockdown of METTL3 significantly reduced the m⁶A level of FOXO3 mRNA (Fig 4M) and reduced the binding of YTHDF1 to FOXO3 mRNA (Fig 4N). Rescue with wild-type METTL3 in METTL3-knockdown HCC cells significantly restored the binding of FOXO3 mRNA to YTHDF1, while a rescue using mutant METTL3 only showed a slight increase in binding compared to the control (Appendix Fig S4I), suggesting that METTL3-mediated m⁶A methylation at the 3′UTR of FOXO3 promoted FOXO3 mRNA stability in an YTHDF1-dependent manner. Moreover, by analyzing the dataset of liver tumors, we found a significant positive correlation between YTHDF1 and FOXO3 (coefficient = 0.327, $P = 0.00164$) (Fig 4O), and between METTL3 and FOXO3 (coefficient = 0.387, $P = 0.00017$) (Fig 4P) at the mRNA level, respectively. Collectively, we confirmed that FOXO3 was the crucial target of METTL3-mediated m⁶A modification in HCC.

## Overexpression of FOXO3 rescued the m⁶A-depedent sorafenib sensitivity in HCC by inhibiting autophagy

To scrutinize the connectivity among the direct targets by FOXO3 in the autophagy signaling pathway, we analyzed two databases by which FOXO3 was induced by doxycycline (Appendix Fig S5A and B). Overexpression of FOXO3 down-regulated gene expression profiling in autophagy, while in SMMC-7721 cells, our results showed that knockdown of FOXO3 induced the transcription of autophagy-related genes, including ATG3, ATG5, ATG7, ATG12, ATG16L1, and MAP1LC3B (Fig 5A). Consistent with down-regulation of METTL3, FOXO3 was dramatically down-regulated in sorafenib-resistant HepG-2 cells (Fig 5B). Knockdown of FOXO3 by two independent shRNAs increased autophagic activity in three HCC cell lines as indicated by LC3-II expression level (Fig 5C, Appendix Fig S5C and D). To determine whether FOXO3 played a crucial role in METTL3-dependent autophagy, we overexpressed FOXO3 in sorafenib-

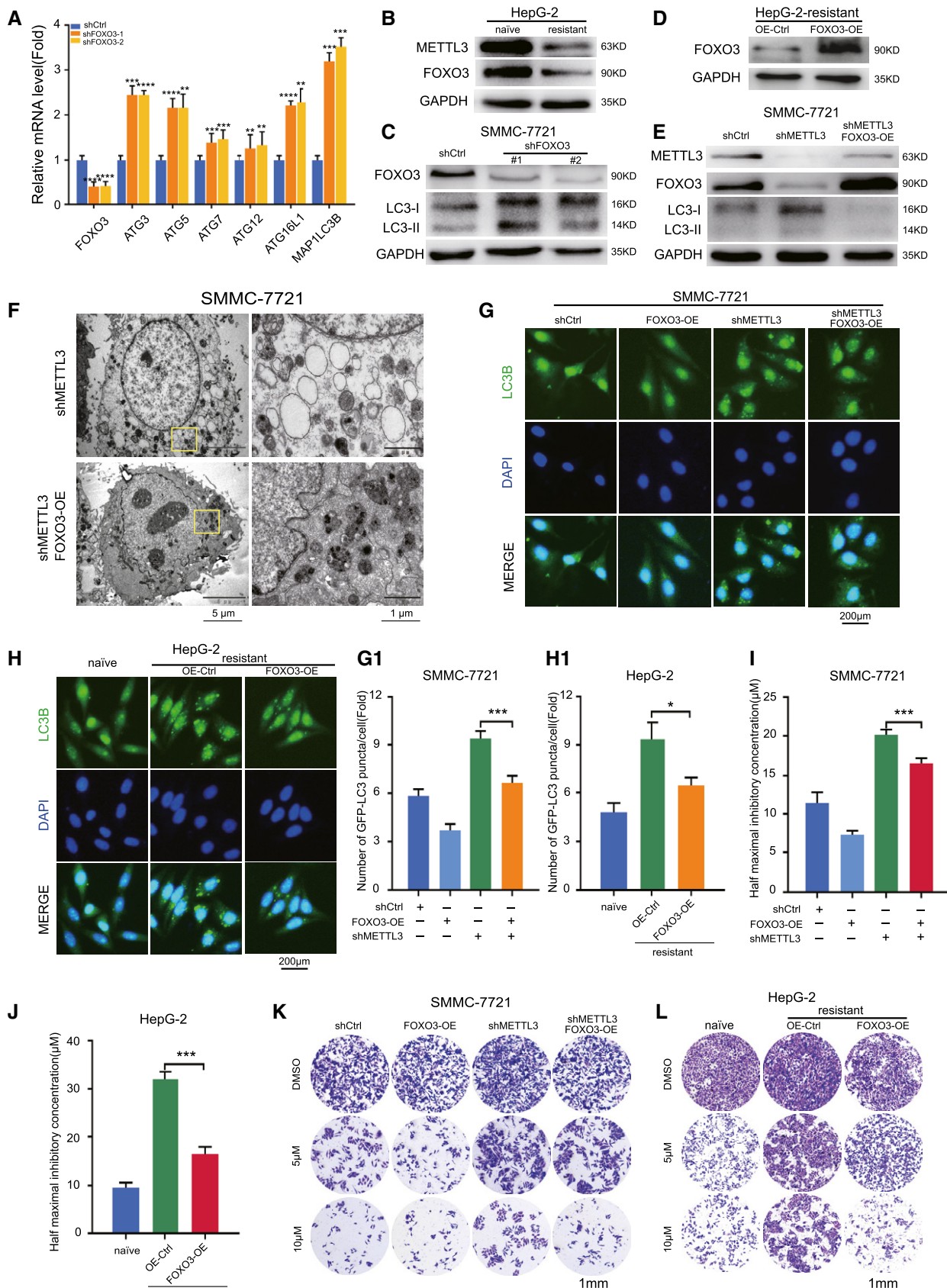

Figure 5.

**Figure 5.  Overexpression of FOXO3 rescued the m$^6$A-dependent sorafenib sensitivity under hypoxia by inhibiting autophagy.**

A  Validation of biomarker genes related to autophagy in FOXO3-knockdown Bel-7402 cells by RT–PCR.
B  Protein level of FOXO3 in naïve HepG-2 and sorafenib-resistant HepG-2 cells under hypoxia for 48 h.
C  Protein levels of LC3 I/II in FOXO3-knockdown SMMC-7721 cells.
D  Overexpression of FOXO3 in sorafenib-resistant HepG-2 cells by Western blot.
E  Protein levels of LC3 I/II in METTL3-knockdown and FOXO3-overexpression SMMC-7721 cells under hypoxia for 48 h.
F  Electron micrographs of METTL3-knockdown SMMC7721 cells and METTL3-knockdown and FOXO3-overexpression SMMC7721 cells under hypoxia for 48 h.
G  Representative immunostaining images of LC3 in METTL3-knockdown SMMC7721 cells and METTL3-knockdown and FOXO3-overexpression SMMC7721 cells under hypoxia for 48 h. Scale bar, 200 μm. The quantification of the numbers of GFP-LC3 puncta/cell by Imaris in G-1.
H  Representative immunostaining images of LC3 in naïve HepG-2 cells and sorafenib-resistant HepG-2 cells under hypoxia for 48 h. Scale bar, 200 μm. The quantification of the numbers of GFP-LC3 puncta/cell by Imaris in H-1.
I  The IC50 of METTL3-knockdown SMMC7721 cells with overexpressing FOXO3 after treated with sorafenib for 24 h under hypoxic condition.
J  The IC50 of naïve HepG-2 cells and sorafenib-resistant HepG-2 cells with overexpressing FOXO3 after treated with sorafenib for 24 h under hypoxic condition.
K  Cell survival assay of METTL3-knockdown SMMC7721 cells with overexpressing FOXO3 after treated with sorafenib for 24 h under hypoxic condition.
L  Cell survival assay of naïve HepG-2 cells and sorafenib-resistant HepG-2 cells with overexpressing FOXO3 after treated with sorafenib for 24 h under hypoxic condition.

Data information: In all relevant panels, *$P < 0.05$; **$P < 0.01$; ***$P < 0.001$; ****$P < 0.0001$; two-tailed *t*-test. Data are presented as mean $\pm$ SD and are representative of 3 independent experiments.
Source data are available online for this figure.

resistant HepG-2 cells (Fig 5D) and METTL3-knockdown SMMC-7721 cells (Fig 5E, Appendix Fig S5E and F). Our results showed that overexpression of FOXO3 in METTL3-knockdown cells decreased autophagic activity with lipidation of LC3 level mediated by METTL3 (Fig 5E), reduced the number of autophagosomes (Fig 5F), and retarded the subcellular redistribution of GFP-LC3 (Fig 5G). Similarly, overexpression of FOXO3 in sorafenib-resistant HepG-2 cells showed the consistent results by fluorescent immunostaining (Fig 5H). Further, we determined whether overexpression of FOXO3 could rescue the m$^6$A-mediated sorafenib sensitivity in HCC cells. We found that overexpression of FOXO3 significantly sensitized the m$^6$A-mediated resistant HCC cells to sorafenib treatment either by reducing their IC50 values (Fig 5I and J) or by using cell survival assays (Fig 5K and L). Taken together, these results suggested that FOXO3 played a critical role in METTL3 depletion-mediated sorafenib resistance in HCC.

## METTL3 depletion significantly enhanced sorafenib resistance via the METTL3/FOXO3 axis *in vivo*

To verify the role of METTL3 in mediating sorafenib resistance in HCC progression, we first performed a syngeneic xenograft mouse model in C57BL/6 mice with Hepa1-6 engineered with a METTL3 knockout by CRISPR/Cas9 (Appendix Fig S5G) and drug administration with two regimens: DMSO and sorafenib (50 mg/kg every 2 days, intraperitoneally). Consistent with our previous *in vitro* results, knockout of METTL3 significantly abolished the inhibitory effects of sorafenib treatment *in vivo*, as indicated with the increased tumor size and tumor weight (Fig 6A–C). Furthermore, we established an orthotopic liver tumor xenograft mouse model by injection of Bel-7402 cells directly into the liver parenchyma of NOD-SCID mice. Knockdown of METTL3 abolished the tumor suppressive effects of sorafenib in the liver tumor microenvironment compared to the control group (Fig 6D and E). To further substantiate the role of the METTL3/FOXO3 axis in liver cancer *in vivo*, we performed a syngeneic xenograft mouse model by establishing a subcutaneous implantation in C57BL/6 mice with Hepa1-6 engineered with METTL3 knockdown or FOXO3 overexpression (Appendix Fig S5H) and started drug administration as previously. Consistently,

knockdown of METTL3 by shRNA significantly abolished the inhibitory effects of sorafenib treatment *in vivo*. Notably, overexpression of FOXO3 in the METTL3-knockdown Hepa1-6 cells rescued the sorafenib sensitivity effect, as reflected by the significant decrease of tumor size and tumor weight comparing to the groups treated with solvents (Fig 6F–H). Overexpression of FOXO3 in the METTL3-knockdown cells under treatment with sorafenib showed decreased proliferation, autophagy and angiogenesis compared to METTL3 depletion alone, as indicated by decreased Ki67 (marker for cell proliferation), LC3-B (marker for autophagy), and VEGF-A (marker for angiogenesis), respectively (Fig 6I–L).

In addition, we performed two HCC patient-derived xenograft (PDX) models in BALB/C nude mice and administrated with four regimens (Fig 7A and Table EV3). Notably, depletion of METTL3 by using *in vivo*-optimized RNAi significantly rescued the xenograft tumor growth compared with the control under sorafenib treatment (Fig 7B–D and I), which showed increased proliferation, autophagy, and angiogenesis (Fig 7E) as indicated by Ki67 (Fig 7F), LC3-B (Fig 7G), and VEGF-A (Fig 7H), respectively. Taken together, we concluded that METTL3 depletion significantly enhanced sorafenib resistance mainly via the METTL3/FOXO3 axis *in vivo*.

## Discussion

Dysregulation of epigenetic modifications has been shown as a common feature of most human cancers. Apart from DNA and histone modifications, RNA m$^6$A modification has been recently proposed to be another important layer of epigenetic regulation in a variety of cellular processes such RNA splicing, protein translation, and stem cell renewal (Wang *et al*, 2014, 2015). Alteration of epigenetic m$^6$A machinery is often associated with cancer development and progression (Niu *et al*, 2018). However, the role of the m$^6$A modification in various physiological events, such as chemical stimulus response, has not yet been fully understood. Currently, occurrence of resistance to chemotherapy and targeted therapies is a major obstacle in cancer treatment. Yan *et al* first identified the dynamic m$^6$A methylome as an additional epigenetic driver for reversible TKI (tyrosine kinase inhibitor) tolerance (Yan *et al*,

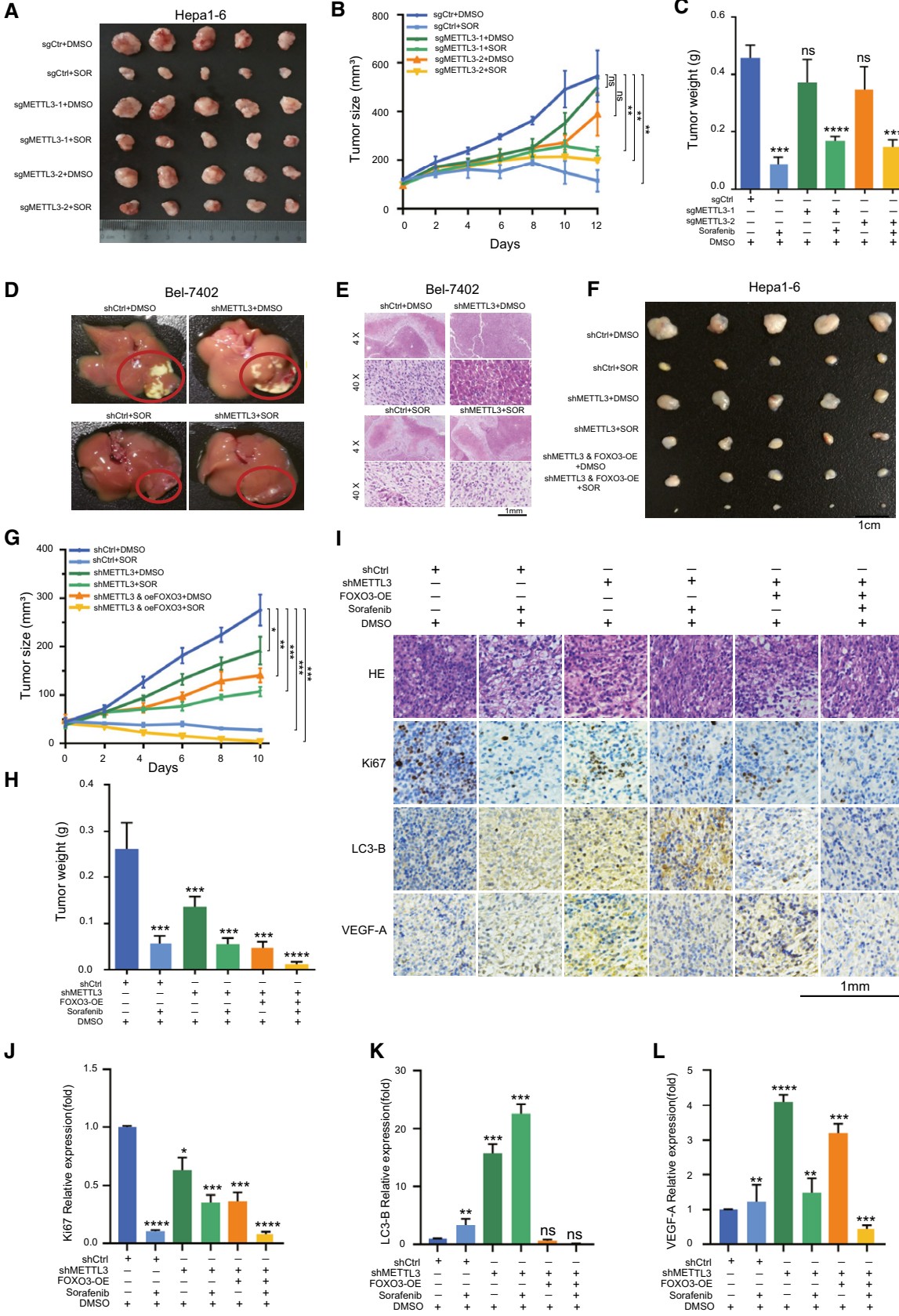

**Figure 6.**

**Figure 6. METTL3 depletion significantly enhanced sorafenib resistance via the METTL3/FOXO3 axis *in vivo*.**

A  Depletion of METTL3 in Hepa1-6 cells by CRISPR-Cas9 knockout enhanced sorafenib resistance *in vivo*.

B, C Growth curve (B) and tumor weight (C) of Hepa1-6 cell-derived xenografts in the subcutaneous implantation mouse model in six treatment groups for 12 days (*n* = 5).

D  Knockdown of METTL3 slightly increased liver tumor growth treated with sorafenib in orthotopic xenograft mouse model. Stable METTL3-knockdown Bel-7402 cells and control cells were injected into the liver of each NOD/SCID mouse. Three weeks after injection, livers were separated for pathological analysis. Red circle indicates tumors in the livers.

E  Liver tumors in orthotopic xenograft mouse model were confirmed by hematoxylin and eosin staining.

F  Overexpression of FOXO3 rescued sorafenib resistance mediated by METTL3 knockdown in xenograft liver tumors. Representative xenograft tumors at endpoint, containing 6 groups.

G, H Growth curve (G) and tumor weight (H) of Hepa1-6 cell-derived xenografts in the subcutaneous implantation mouse model in 6 treatment groups for 10 days (*n* = 5).

I  H&E and immunohistochemical images of Ki67, LC3-B, and VEGF-A in Hepa1-6 cell-derived xenografts in 6 treatment groups.

J–L Quantification of Ki67 (K), LC3-B (K), and VEGF-A (L) expression in immunohistochemical images by Image Pro Plus (IPP) analysis.

Data information: In all relevant panels, $*P < 0.05$; $**P < 0.01$; $***P < 0.001$; $****P < 0.0001$; two-tailed *t*-test. Data are presented as mean $\pm$ SD.

---

2018). They found that the FTO-mediated $m^6A$ demethylation promoted mRNA stability of some transcripts related to proliferation and survival such as MERTK and BCL-2, to regulate development and maintenance of TKI resistance (Yan *et al*, 2018). For HCC, few studies have focused on the role of $m^6A$ modification in regulating drug resistance.

In this study, we were the first to demonstrate that down-regulation of METTL3 was significantly associated with sorafenib resistance in HCC with a series of transcriptome sequencing analyses. We further found that METTL3 depletion-mediated $m^6A$ level suppression promoted resistance of several HCC cell lines to sorafenib by activating the autophagy signaling pathway. Rescue experiment using shRNA-resistant wild-type METTL3, but not catalytic mutant METTL3, in METTL3-depleted HCC cells and in sorafenib-resistant HCC cells restored their sensitivity to sorafenib under hypoxia, indicating the director role of METL3-mediated $m^6A$ modification in sorafenib resistance of HCC. Recently, Song *et al* reported a negative role of METTL3 in modulating autophagy in ischemic heart disease (Song *et al*, 2019). Our data also suggested a similar modulatory role of METTL3 in regulating autophagy, and we found that METTL3-dependent sorafenib resistance in HCC was mediated by promoting autophagy signaling pathway. Previous studies have shown that autophagy acts as a protective mechanism in cancer cells exposed to a variety of anticancer drugs (Doherty & Baehrecke, 2018). Knockdown of METTL3 expression increased the number of autophagosomes and LC3 accumulation in HCC cells, indicating suppression of autophagy may reverse the $m^6A$-dependent resistance to sorafenib, a molecular targeted drug that inhibits RAF/MEK/ERK pathway and VEGFR/PDGFR pathway. Previous studies have reported that combination of anti-ANXA3 (Tong *et al*, 2018) or Cdk5 inhibition (Ardelt *et al*, 2019) with sorafenib acts as novel strategies to improve sorafenib response in HCC treatment. We applied combination of autophagy inhibitor with sorafenib for HCC treatment under hypoxia and obtained the significant sensitivity effects of the cells to sorafenib, suggesting another novel strategy for sorafenib-resistant HCC treatment.

To elucidate the mechanism by which METTL3-mediated $m^6A$ modification regulated autophagy, we applied a serial of RNA-seq and $m^6A$-Seq to identify the downstream target of METTL3-mediated $m^6A$ methylation. Based on the enrichment analysis, we identified FOXO3, a crucial transcription factor in multiple oncogenic pathways including ERK16, NF-κB, and PI3k-AKT signal cascades, as a primary regulator connecting $m^6A$ modification with autophagy. FOXO3 was previously shown to function as a tumor suppressor regulating cell cycle arrest (Yusuf *et al*, 2004), apoptosis (Jonsson *et al*, 2005), and autophagy (Mammucari *et al*, 2007; Zhou *et al*, 2012; Gomez-Puerto *et al*, 2016). In addition, members of the FOXO family, such as FOXO1, FOXO3, FOXO4, and FOXO6 (Jacobs *et al*, 2003), have been found to participate in various physiological responses, such as DNA damage (Tran *et al*, 2002), caloric restriction (Carrano *et al*, 2009) and oxidative stress (Essers *et al*, 2005), and play crucial roles in cancer initiation, progression, and chemoresistance. Interestingly, previous reports have shown that FOXO3 up-regulates autophagy in other tissues, including skeletal muscle (Mammucari *et al*, 2007) and bone (Gomez-Puerto *et al*, 2016). For example, Mammucari *et al* reported that FOXO3 controls the transcription of autophagy-marker genes, such as LC3 and BNIP3, and BNIP3 was the main player to mediate the effect of FOXO3 on autophagy in skeletal muscle (Mammucari *et al*, 2007). Gomez-Peurto *et al* showed that FOXO3 induces autophagy in human mesenchymal stem cells in bone upon elevated ROS levels (Gomez-Puerto *et al*, 2016). In contrast, Schaffner *et al* recently showed that FOXO3 controls autophagy in neuronal development, and FOXO3 deficiency induced autophagy to correct abnormal dendrite and spine development (Schaffner *et al*, 2018), which is consistent with our results that down-regulation of FOXO3 triggers autophagy signaling pathway in liver tumor tissue under intratumoral environment-like conditions. These reports raise controversial debates on the role of FOXO3 in autophagy in various tissues and physiological conditions.

In this study, we validated FOXO3 as a direct target of METTL3 in HCC under intratumoral environment-like conditions. Silencing of METTL3 dramatically reduced FOXO3 expression at both RNA and protein levels. METTL3 directly mediated $m^6A$ methylation at the 3′UTR of FOXO3 and enhanced its mRNA stability, while mutation of the $m^6A$ site (RRACH) replacing adenosine with thymine completely abolished $m^6A$ modification. By using MeRIP assay, we confirmed the YTHDF1 binding site at the 3′UTR of FOXO3. A previous study showed that YTHDF1, as a major $m^6A$ reader protein, actively stabilizes $m^6A$-labeled RNA and promotes protein synthesis (Wang *et al*, 2015). Consistently, we found that knockdown of YTHDF1 significantly reduced the expression level of FOXO3; however, mutation of the $m^6A$ site abolished its reduction, indicating that METTL3-mediated $m^6A$ modification promoted FOXO3 RNA stability in an YTHDF1-dependent manner. The binding of YTHDF1 to FOXO3 mRNA was impaired in METTL3-knockdown HCCs;

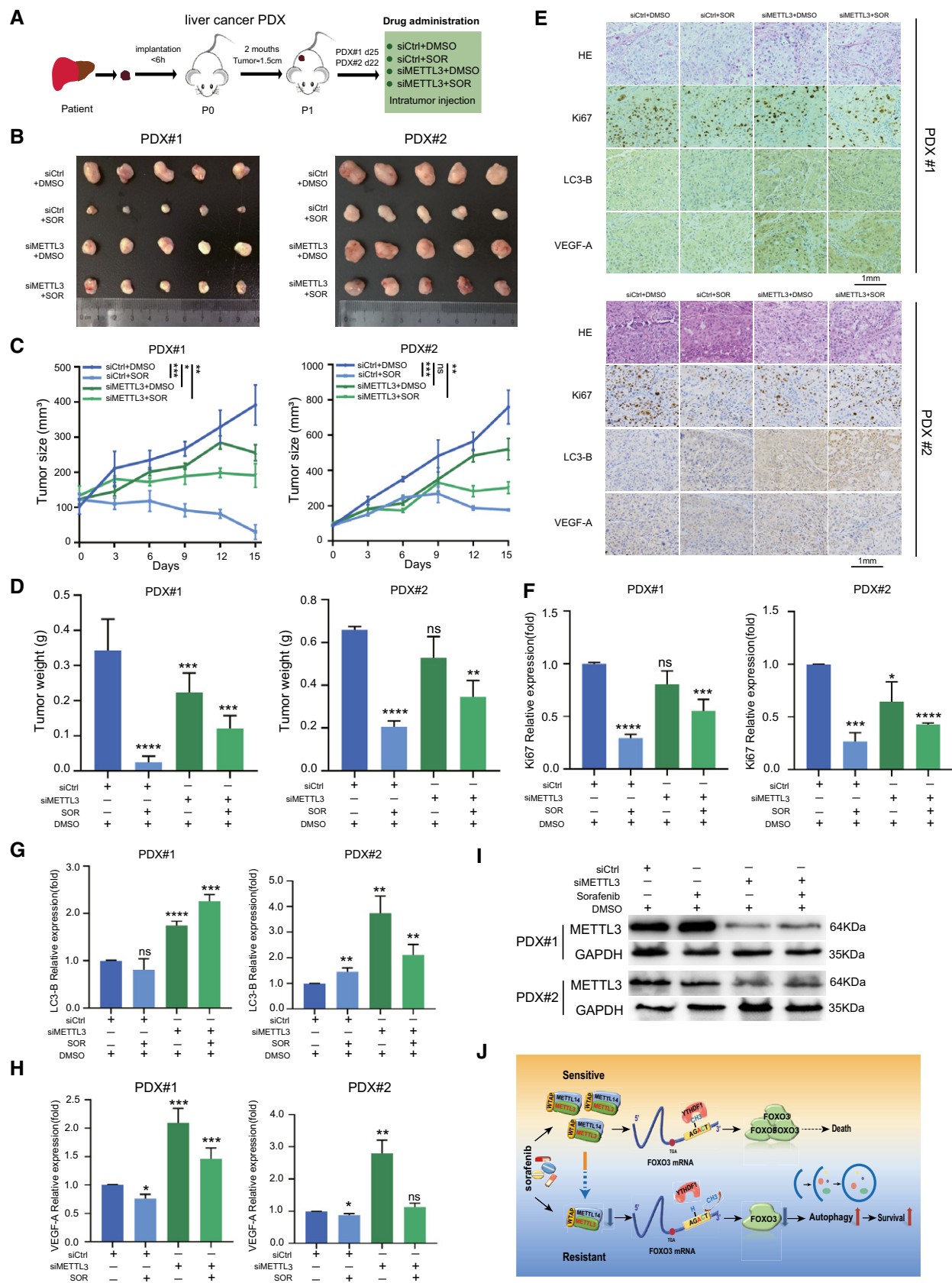

**Figure 7.**

**Figure 7. METTL3 depletion significantly enhanced sorafenib resistance in HCC patient-derived xenografts.**

A    Graphic illustration of two human liver cancer PDXs mouse model based sorafenib (SOR) treatment regimen.

B–D  *In vivo* analyses of tumor (B), growth (C), and weight (D) in mice that were subcutaneously implanted with tumor tissues from two human liver cancer patients and divided into four groups.

E    H&E and immunohistochemical images of Ki67, LC3-B, and VEGF-A in randomly selected PDX#1 and PDX#2 tumors.

F–H  Quantification of Ki67 (F), LC3-B (G), and VEGF-A (H) expression in immunohistochemical images from PDX tumors by Image Pro Plus (IPP) analysis.

I    Knockdown of METTL3 by siRNA in PDX#1 and PDX#2 tissues was confirmed by Western blot.

J    Working model. The molecular mechanisms of $m^6A$-dependent sorafenib resistance in liver cancer. The expression of METTL3 was down-regulated in sorafenib-resistant liver cancer and caused decrease of FOXO3 by the $m^6A$ modification in an YTHDF1-dependent manner. Degradation of FOXO3 thus promoted autophagy-associated sorafenib resistance in liver cancer.

Data information: In all relevant panels, $*P < 0.05$; $**P < 0.01$; $***P < 0.001$; $****P < 0.0001$; two-tailed $t$-test. Data are presented as mean $\pm$ SD.

however, rescue with wild-type METTL3, but not the catalytic mutant of METTL3, restored the binding capacity of YTHDF1 to FOXO3 mRNA. Our results thus provide direct evidence of the involvement of $m^6A$ in the regulation of FOXO3 via YTHDF1 binding. Furthermore, we investigated whether FOXO3 was the key regulator for the $m^6A$-induced autophagy. To do so, we overexpressed FOXO3 in METTL3-knockdown HCC cells and analyzed autophagy-related gene expression. Here, we found that, as a crucial transcription factor, FOXO3 directly negatively regulates the expression of ATG3, ATG5, ATG7, ATG12, ATG16L1, and MAP1LC3B in HCC, while knockdown of FOXO3 conversely induced their transcription activity. Our results suggest that FOXO3 plays a crucial role in mediating the $m^6A$-dependent chemo-sensitivity in HCC by inhibiting autophagy signaling pathway.

Previously, another group has shown that the $m^6A$ modification was involved in promoting liver cancer progression through YTHDF2-mediated SOCS2 degradation (Chen *et al*, 2018). Nonetheless, our study mainly focused on the role of METTL3-mediated $m^6A$ modification in sorafenib resistance under intratumoral environment-like conditions and identified that the METTL3/FOXO3 axis is involved in acquired sorafenib resistance in HCC. To further investigate the potential manifestation of the METTL3/FOXO3 axis in the *in vivo* model of HCC with sorafenib treatment, we have used four different mouse models, including the subcutaneous tumor mouse model, the orthotopic liver tumor xenograft mouse model, the human liver cancer PDX model, and the syngeneic xenograft mouse model to mimic the liver microenvironment and to examine the effects of METTL3 depletion on sorafenib treatment. Our results showed that the decreased METTL3-$m^6A$ function enhanced autophagy and angiogenesis in HCC through the METTL3/FOXO3 axis, leading to establishment of resistant phenotypes. In addition, our pathological analysis confirmed the molecular mechanism identified in our previous *in vitro* experiment, by which the stable knockdown of METTL3 increased expression of Ki67, LC3-B, and VEGF-A in HCC cells. These results suggest that the heterogeneous METTL3-deletion $m^6A$ axis allows a subpopulation of cells to rapidly prevail their $m^6A$ methylome with related genes to withstand an initial onslaught of sorafenib.

In summary, we uncovered an important role of METTL3-mediated $m^6A$ modification in regulating HCC sorafenib resistance under intratumoral environment-like conditions (Fig 7J). The expression of METTL3 was down-regulated in sorafenib-resistant liver tumors and caused degradation of FOXO3, which promoted autophagy-induced sorafenib resistance in HCC. Our study demonstrated that the METTL3/FOXO3 axis acts as a critical regulator in sorafenib-resistant HCC cells, linking the intratumoral

microenvironment to autophagy. More importantly, our study provides a novel therapeutic strategy to enhance the treatment response in sorafenib-resistant HCC.

# Materials and Methods

### Tissue culture

The HCC tumors and sorafenib-resistant tumors were obtained at the Third Affiliated Hospital of Sun Yat-Sen University and Southern Medical University and were approved by the institutional review board of the hospitals. The study is compliant with all relevant ethical regulations regarding research involving human participants. HepG-2, Hepa1-6, HEK-293T, WRL68 and HUVEC cell lines were obtained from the American Type Culture Collection (ATCC). SMMC-7721, and Bel-7402 cell lines were obtained from Procell. SMMC-7721, Bel-7402, HepG-2, WRL68, and HEK-293T cells were cultured in Dulbecco's modified Eagle's medium (DMEM, Corning, USA) with 10% FBS at 37°C in 5% $CO_2$. Hepa1-6 cells (a liver cancer cell line derived from C57BL/6 mice) were cultured in DMEM with 10% FBS and 100 U/ml Pyruvic Acid Sodium. HUVECs were cultured in endothelial cell medium (ECM, Gibco). All cell lines were stored in multiple backup upon receipt to reduce risk of phonotypic drift and tested with mycoplasma free by Mycoplasma Stain Assay Kit (C0296, Beyotime, China). Cell line authentication was validated by the STR analysis using GenePrint 10 System according to the manufacturer's instruction (B9510, Promega, USA). Culture in hypoxic condition was set in a low oxygen incubator equilibrated with certified gas containing 1% $O_2$, 5% $CO_2$, and 94% $N_2$. Sorafenib-resistant HepG2 cell line was generated by exposing the cells with sorafenib at 5% of IC concentration for 3 days and gradually increased the concentration by 5% of IC till reaching to the IC50.

### Plasmid constructions, cell transfection, and virus infection

Stable knockdown of target genes was accomplished by lentiviral-based specific short-hairpin RNA (shRNA) delivery. Stable knockout of target genes was accomplished by lentiCRISPRv2 delivery. The vector pLKO.1-puromycin or pLKO.1-hygromycin was constructed by using primers listed in Appendix Table S1. For FOXO3 overexpressing constructs, human FOXO3 cDNA (NM_201559) and mouse FOXO3 (NM_019740) were cloned into pCDH puro lentiviral vector (CD510B-1, System Biosciences). For stable knockdown assay, pLKO.1 vector together with packing and helper plasmids PAX2 and MD2G were co-transfected into HEK-293T cells by Calcium

Phosphate Transfection Kit (CAPHOS-1KT, Sigma). Viruses were produced, filtered, and titrated according to the instruction, and infected the desired cells with 8 μg/ml polybrene (TR-1003, Sigma). After screening by puromycin (1 μg/ml) or hygromycin (100 μg/ml) in corresponding concentration for 3–5 days, the stable knockdown or overexpression cells were harvested.

### Antibodies

Anti-HIF-1α (WB 1:500; IHC 1:100; ab51608), anti-METTL3 (WB 1:1,000; IHC 1:500; ab195352), anti-FOXO3 (WB 1:1,000; ab53287), goat anti-rabbit IgG (WB 1:2,000; ab6721), and goat anti-rabbit IgG-FITC (IF 1:200, ab6717) antibodies were purchased from Abcam, USA. Anti-MAP1LC3-B (WB 1:1,000; IHC 1:100; IF 1:100; A7198), anti-PDGF-B (WB 1:1,000; A1195), anti-VEGF-A (WB 1:1,000; IHC 1:100; A5708), anti-Ki67 (IHC 1:100; A11390), anti-GAPDH (WB 1:1,000; AC027), anti-tubulin (WB 1:5,000; AC021), anti-β-actin (WB 1:50,000; AC026), anti-YTHDF1 (IHC 1:1,000; A18126), and anti-YTHDF2 (IHC 1:1,000; A15616) antibodies were purchased from Abclonal, China. Anti-m$^6$A (MeRIP 1:1,000; ABE572) antibody was purchased from Merck Millipore (Massachusetts, USA). Western blot analyses were performed by standard methods described previously (Niu et al, 2019).

### Quantitative RT–PCR

Total RNA was isolated using TRIzol reagent according to the manufacturer's instruction (Thermo Fisher, USA), and cDNA was synthesized by using the PrimeScript RT Reagent Kit (RR036A, Takara, Japan). The resulting cDNA was used for quantitative RT–PCR by using SYBR Green Master Mix (RR820B, Takara, Japan) in 7500 apparatus (Applied Biosystems). RT–PCR primer sequences are listed in Table EV1.

### Immunohistochemistry

For immunohistochemical (IHC) analysis, HCC tissue slides were deparaffinized and rehydrated through an alcohol series followed by antigen retrieval with sodium citrate buffer. Tumor sections were blocked with 5% normal goat serum (Vector) with 0.1% Triton X-100 and 3% H$_2$O$_2$ in PBS for 60 min at room temperature and then incubated with appropriate primary antibodies 4°C overnight. IHC staining was performed with horseradish peroxidase (HRP) conjugates using DAB detection. Nuclei were counterstained with Hoechst. Images were taken with Nikon (Japan).

### Immunofluorescence and transmission electron microscopy

For immunofluorescent staining, HCC cells with or without shMETTL3 were treated with hypoxia for 24 h. Then, cell slides were incubated with different primary antibodies and were subsequently incubated with secondary antibodies. Nuclei were counterstained with DAPI. Images were obtained by Nikon (Japan). For TEM assay, cells were fixed with 3% glutaraldehyde in 0.1 M phosphate buffer (pH 7.4), followed by the fixation with 1% OsO$_4$. After dehydration, 10-nm thin sections were prepared and stained with uranyl acetate and plumbous nitrate before examined under a JEM-1230 transmission electron microscope (JEOL, Tokyo, Japan). High-resolution digital images were acquired from a randomly selected 10 different fields for samples of each condition.

### Half maximal inhibitory concentration assay (IC50)

Cells were cultured in 96-well plates with fresh medium in $1.5 \times 10^4$ per well. The corresponding concentrations of drug were given to cells for 24 h after the cultured plates were placed in a hypoxic condition (1% O$_2$) for 24 h. After 48 h under hypoxia, Cell Counting Kit-8 (Dojindo, Japan) was used to measure drug sensitivity at 450 nm using a microplate reader (Thermo Fisher, USA) after incubating at 37°C for additional 2–4 h.

### Cell survival assay and clonogenic assays

HCC cells were seeded into 12-well plates at a density of $1.5 \times 10^5$ cells with fresh cell culture medium. Then, plates with cells were treated with related chemo-drugs after 24 h in 1% O$_2$ hypoxic conditions. After another 24 h with medication, the plates were treated with polyformaldehyde fixing solution and crystal violet staining solution (Beyotime, China). The colony pictures were captured with Nikon (Japan).

For CCK8 assay, cells were seeded at 1,000 cells per well in 96-well plates with fresh medium. Cell viability was assayed using Cell Counting Kit-8 (CK04, Dojindo, Japan) at the time in 0, 48, 72, and 96 h. The microplates were incubated at 37°C for additional 2 h. Absorbance was read at 450 nm using a microplate reader (Thermo Fisher, USA).

For clonogenic assay, cells were seeded into 6-well plates at a density of 5,000 cells with fresh cell culture medium and maintain the related drug concentration for 7 days. Then, the plates were treated with polyformaldehyde fixing solution and crystal violet staining solution (Beyotime, China) and the colony pictures were captured separately.

### HUVEC tube formation assay

The tumor-conditioned medium (TCM) from SMMC-7721, Bel-7402, and HepG-2 cells cultured under hypoxic conditions for 48 h was collected and added to 48-well plates, which were coated with Matrigel (Corning, USA) and seeded with HUVECs after 6-h serum starve treatment; after 3 h of incubation at 37°C, the cells were observed under a microscope to detect the formation of capillary-like structures, and the number of tubes was quantified with ImageJ.

### RNA m$^6$A dot blot assay

The 300 ng poly(A) RNAs were gradient-diluted and spotted onto a nylon membrane. The membranes were UV (245 nm)-crosslinked, blocked, incubated with m$^6$A antibody and horseradish peroxidase-conjugated anti-rabbit immunoglobulin G, and subsequently washed with PBST and substrated in visualizer (4600, Tanon, China). The same gradient-diluted total RNAs were spotted on the membrane, stained with 0.2% methylene blue in 0.3 M sodium acetate (pH 5.2) for 2 h, and washed with ribonuclease-free water for 1 h.

## m⁶A sequencing and m⁶A-RNA immunoprecipitation assay

MeRIP sequencing was performed by standard methods described previously (Niu *et al*, 2019). In brief, total RNAs were isolated from SMMC-7721 cells treated with normoxic or hypoxic conditions and chemically fragmented into around 100 nt. The fragmented RNA was incubated with m⁶A antibody for immunoprecipitation according to the manufacture's instruction (MeRIP m⁶A Kit, Merck Millipore, USA). Enrichment of m⁶A containing mRNA was sent for high-throughput NGS and validated by quantitative RT–PCR. For NGS, purified RNA fragments from m⁶A-MeRIP were used for library construction according to the manufacture's instruction (NEB Next Ultra RNA library Prep Kit, Illumina) and sequenced with Illumina Hi-Seq 2000. Library preparation and high-throughput sequencing were performed by Novogene (Guangzhou, China). Sequencing reads were aligned to the human genome GRCh37/hg19 by Bowtie2, and the m⁶A peaks were detected by magnetic cell sorting as described.

## Luciferase reporter assays and mutagenesis assay

The 3′UTR sequence of FOXO3 (NM_001455) was amplified by PCR using the genomic DNA from Bel-7402 cells and subcloned into the dual-luciferase vector pmiGLO (C838A, Promega, USA). A putative m⁶A recognition site was identified in 3′UTR. Mutagenesis from A to T was generated by Quick Change II Site-Directed Mutagenesis Kit (200523, Agilent, USA) according to the manufacture's instruction. Luciferase activity was measured by Dual-Luciferase Reporter Gene Assay Kit (RG028, Beyotime, China) in GM2000 (Promega). Experiments were performed in triplicates. The firefly luciferase activity values were normalized to the Renilla luciferase activity values that reflect expression efficiency.

## Animal experiments

All animal experiments were approved by the Animal Ethics Committee of Sun Yat-Sen University (No. [2017]03-07). For the subcutaneous tumor models, $1 \times 10^6$ SMMC-7721 cells were injected into 5-week-old male BALB/C nude mice, and the tumor tissues were taken out 1 month later for future experiments. For the drug-resistant subcutaneous tumor models, sgMETTL3-1 and sgMETTL3-2 stable Hepa1-6 cells were injected subcutaneously into either flank of 5-week-old male C57BL/6 mice. Drug administration was adopted when the tumors reached about 50 mm³ in size, at which point mice were randomized for treatment with DMSO(intraperitoneally) or sorafenib (50 mg/kg/every 2 days, intraperitoneally). For the drug-resistant subcutaneous tumor models, $1 \times 10^6$ control, shMETTL3, shMETTL3&FOXO3-OE stable Hepa1-6 cells were resuspended in 100 μl PBS and injected subcutaneously into either flank of 5-week-old male C57BL/6 mice. Tumor volumes were measured using a vernier caliper and calculated using the following formula: volume (cm³) = L×W²/2 with L and W representing the largest and smallest diameters, respectively. Drug administration was adopted when the tumors reached about 50 mm³ in size, at which point mice were randomized for treatment with DMSO(intraperitoneally) or sorafenib (50 mg/kg/every 2 days, intraperitoneally). For liver orthotopic xenograft mouse model, $1 \times 10^6$ shCtrl or shMETTL3 Bel-7402 cells were resuspended in

100% Matrigel (BD Biocoat, Corning) and injected into the left lobes of the livers in 5- to 6-week-old male NOD/SCID mice, and then, the incision was closed using surgery suture threads with needle and medical adhesive bandage. Drug administration began a week after inoculation (30 mg/kg/day, intraperitoneally). At three weeks post-inoculation, tumor-bearing mice were sacrificed and livers were harvested for histological analysis. For the patient-derived tumor xenograft model, PDX tumors were generated from liver cancer patients. Human liver tumor tissue was implanted within 6 h, expanded and sub-implanted into 5-week-old male BALB/C nude mice. Tumor size was measured weekly using a caliper, and tumor volume was calculated using the standard formula above. Drug administration began 4 weeks after tumors reached about 100 mm³ in size with sorafenib (50 mg/kg/every 3 days, intraperitoneally) or siCtrl/siMETTL3 intratumor injection. At 2 weeks post-inoculation, tumor-bearing mice were sacrificed. Livers were harvested for histological analysis.

## Bioinformatics analysis

The gene expression profile dataset was downloaded from Gene Expression Omnibus (GEO) database and subsequently analyzed by R (Version 3.4, http://www.bioconductor.org) with edgeR package. Fold change (FC) of gene expression was calculated with a cut-off of $\log_2 FC \geq 1$, $\log_2 FC < -1$, and $P$ value < 0.05. The online database about R2, a website of Genomics Analysis and Visualization Platform (https://hgserver1.amc.nl/cgi-bin/r2/main.cgi), was applied to determine the clinical survival of the related genes. KEGG pathway enrichment analysis was performed to investigate the activated pathways of the differentially expressed genes, by applying online tool of DAVID Bioinformatics Resources 6.8 (http://david.ncifcrf.gov) and KOBAS.3.0 (http://kobas.cbi.pku.edu.cn/anno_iden.php). The Search Tool for the Retrieval of Interacting Genes (STRING) database (V10.5, http://string-db.org) was recruited to predict the potential interaction between FOXO3 and autophagy genes at protein level.

## Statistical analysis

Means and SD were analyzed using GraphPad prism 8.0. Two-tailed Student's *t*-test was used to compare the statistical difference between indicated groups. Pearson analysis was used to analyze correlation between genes. Statistical significance was accepted for $P < 0.05$.

# Data availability

The bulk RNA-seq data from this publication have been deposited in NCBI's Gene Expression Omnibus database and assigned the GEO Series accession number GSE143235 (https://www.ncbi.nlm.nih.gov/geo/query/acc.cgi?acc=GSE143235).

**Expanded View** for this article is available online.

## Acknowledgements

We thank Prof. Xiongbin Lu (Indiana University School of Medicine, USA) and Prof. Shengyu Yang (Pennsylvania State University College of Medicine, USA)

for their encouragements, suggestions, discussions, and support covering all aspects on this study. We thank Prof. Wei Zhao (Sun Yat-Sen University) for providing us with wild-type METTL3 and catalytic mutant METTL3 expression vectors. This work was supported in part by grants from the National Natural Science Foundation of China 31701114 (GW), 81701834, 81871994 (WH), 21672266 (XB), 81973359 (XZ); National Engineering Research Center for New Drug and Druggability Evaluation, Seed Program of Guangdong Province, 2017B090903004 (GW); Guangdong Basic and Applied Basic Research Foundation 2019A050510019, 2019B030301005 (GW), 2019B151502063 (WH); Guangzhou Science and Technology Planning Program 201902020018 (WH).

## Author contributors

GW and ZL designed and evaluated the project. ZL, YN, AW, HL, LS, SZ, XC, LC, DC, CC, XZ, XB, and WH performed the experiments and analyzed results. GW and ZL wrote the manuscript, which was critiqued by all authors. GW and WH supervised the study and were responsible for critical revision of the manuscript for important intellectual content.

## Conflict of interest

The authors declare that they have no conflict of interest.

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
