## [Review Process File · The EMBO Journal]

RNA m⁶A methylation regulates sorafenib-resistance in liver cancer through FOXO3-mediated autophagy

Ziyou Lin, Yi Niu, Arabella Wan, Dongshi Chen, Heng Liang, Xijun Chen, Lei Sun, Siyue Zhan, Liutao Chen, Chao Cheng, Xiaolei Zhang, Xianzhang Bu, Weiling He, Guohui Wan

Review timeline:

Submission date:	8 August 2019
Editorial Decision:	27 September 2019
Revision received:	25 December 2019
Editorial Decision:	24 February 2020
Revision received:	13 March 2020
Editorial Decision:	24 March 2020
Revision received:	25 March 2020
Accepted:	3 April 2020

Editor: Stefanie Böhm

Transaction Report:

1st Editorial Decision

27 September 2019

Thank you for submitting your study on the role of m⁶A in sorafenib-resistance of liver cancer for consideration by The EMBO Journal. We have now received three referee reports on your study, which are included below for your information.

As you will see, the reviewers express an overall interest in the study, but they also raise several major concerns. In addition to more specific technical issues that should be addressed, referee #1 and #2 find that further experiments to define the role of m⁶A and YTHDF1 are needed (ref#1 major 5, concern 23 and ref#2 major 3, 4). Referee #1 also points out that more patient-derived materials should be analyzed. We realize that repeating all experiments with patient-derived materials is not within the scope of a revision, but do agree on the need for validation of key findings in another experimental system. In a revised version it will furthermore be crucial to clarify the origin and identity of the patient-derived transcriptomic data (ref#1 concerns 6, 7 and ref#2 major comment 1) and of the PAR-CLIP dataset (ref#1 concern 21 and ref#2 major comment 4).

Should you be able to address the key points raised by the referees, then we would like to invite you to prepare and submit a revised manuscript. Please note that EMBO Journal policy allows only a single round of major revision.

REFeree REPORTS

Referee #1:

The manuscript by Lin et al explores the role of RNA m⁶A modification and m⁶A-methyltransferase METTL3 in sorafenib-resistance phenotype in hepatocellular carcinoma (HCC). The authors observe that sorafenib-resistant tumors have low levels of METTL3 and m⁶A. Authors

report that METTL3 stabilizes FOXO3 mRNA in an m6A- and YTHDF1-dependent manner that sensitizes HCC tumors cells to Sorafenib. Although these findings are interesting and hold potential therapeutic implications, authors must address following concerns before reaching aforementioned conclusions.

Major concerns:

1. The authors overuse cell lines, especially HepG2, which is often cross-contaminated. They use a patient-derived xenograft in a late experiment. As the claims are focused on hypoxia, it would be important to consider using more patient-derived materials and lines to avoid the effects of long term culture in room air.
2. The exclusive focus on loss-of-function studies without METTL3 rescue or CRISPR studies limits confidence. It would be helpful to better develop controls. The in vivo studies use a single shRNA, which is an even lower threshold.
3. I would strongly suggest the use of over expression studies with METTL3 (and loss-of-function with FOXO3) in sorafenib-resistant HCC. The shRNAs for METTL3 have effects separate from sorafenib and METTL3 has been proposed as both an oncogene and tumor suppressor. Is METTL3 both necessary and sufficient?
4. There was no comparison to normal liver cells. Is this a cancer-specific regulation? This would be important to consider as there needs to be consideration of toxicity.
5. The m6A studies are very interesting, but most of the biology is not really linked to m6A itself. It would be valuable to better develop the direct role of m6A in this process.
6. I was surprised that autophagy inhibitors were not used in validation.

Concerns:

- 1) The authors use p-hacking for unbalanced groups in Figures 1H and 7I. This should be corrected.
- 2) The data in Figure 1 is interesting but the clinical aspects should be better developed. The differences in patient response should be distinguished from other patient differences.
- 3) The role of hypoxia is acceptable but somewhat premature. The claim is based on a single subcutaneous result without defining hypoxia or HIF function in the molecular regulation or cell biology. I was not sure why the experiment was not orthotopic, as well.
- 4) All experiments presented in Figure 2 to 5 are performed under hypoxic conditions. These results establish that METTL3-FOXO3 axis regulates autophagy and sorafenib-resistance in HCC cell lines. Are these observations specific to hypoxia only? Authors are encouraged to perform key mechanistic experiments under normoxic conditions as well.
- 5) Authors have used Crystal violet staining to measure cell viability. Since it is a qualitative assay, authors are suggested to perform an additional quantitative cell viability assay as well with appropriate statistics to support their findings.
- 6) On page 4, authors have stated "we performed transcriptome sequencing between liver tumors and sorafenib-resistant tumors to examine the differentially expressed genes". It is not clear in the manuscript whether authors performed an RNA sequencing experiment? If yes then authors need to provide the details of number and type of samples used and also include the protocol in material and methods section. Additionally, it is important for the authors to elaborate the term "liver tumors and sorafenib-resistant tumors".
- 7) If RNA sequencing was not performed, then what is the source of transcriptome data shown in Figure 1? If authors have analyzed a previously available dataset then they should provide the reference as well as describe the method used to characterize sorafenib-resistant tumors from which data was obtained.
- 8) Please better define the "ctrl" for each shRNA experiment.
- 9) In supplementary Figure 1A, authors are suggested to use FDR in place of p-value to plot the data. It seems METTL3 lies on borderline of the cutoff and there are several other genes which are better candidates for follow up study. Authors are suggested to present a better explanation for the selection of METTL3 for the study.
- 10) In Figure 1B, authors are suggested to explain axis label and provide a meaningful axis title.
- 11) In Figure 1D, authors are suggested to provide values on color scale in place of 'min' and 'max'.
- 12) In Figure 1E, authors are suggested to depict the number of tumor samples analyzed in different groups.
- 13) In Figure 1F, it will be useful to include the data points on the Violin plots.
- 14) In Figure 1H, authors should clarify the rationale behind the cutoffs chosen to define 'METTL3 high' and 'METTL3 low' cohorts. Why 'n' numbers are different in two groups?
- 15) In supplementary Figure 1D, 1E and 1F, what is the meaning of "less hypoxia"?
- 16) In Figure 2A, authors are suggested to provide a meaningful label for Y axis with proper units.
- 17) In figure 2, panel D and panel E have been interchanged. Authors are suggested to make the

necessary corrections.

18) Figure 3: 3MA studies are interesting but lack proper controls of 3MA treatment with "ctrl", which I assume is a non-targeting control.

19) On page 6, authors have stated that "Our results revealed that the tumor-conditioned medium (TCM) from HCC cells in the METTL3-knockdown groups showed decreased capacity to stimulate HUVECs tube formation compared to the control groups (Fig 2L and M)". Actually, figure 2L and 2M show an opposite result. Authors are suggested to make the necessary corrections.

20) In figure 4A, authors need to provide a meaningful label for the enrichment plots presented. It's unclear, what these enrichment plots are depicting? On page 7, authors describe these results as "Gene Set Enrichment Analysis (GSEA) was adopted to conduct the gene enrichment pathway in cellular response to stress, which was defined as unfavorable environmental conditions such as hypoxia". Please elaborate.

21) In figure 4C, authors have presented YTHDF1-ParCLIP data and m6A-seq data from SMMC-7721 cells and overlapped it with five other previously available m6A-seq datasets to demonstrate the presence of m6A peaks on FOXO3 mRNA. First, authors have not mentioned the source of YTHDF1-ParCLIP data used in this figure. Did authors perform YTHDF1-ParCLIP experiment? Second, it is not clear whether authors performed an m6A-seq experiment in SMMC-7721 cells. Authors have included a protocol for m6A sequencing in the materials and methods section of the manuscript but it describes that experiment were performed in Bel-7402 and HepG-2 cells but not in SMMC-7721 cells. Please clarify? Also, the m6A seq data obtained in Bel-7402 and HepG-2 cells are not used anywhere in the manuscript.

22) In figure 4E, authors are suggested to present a quantification of immuno-bots and apply proper statistical methods to derive any conclusion about FOXO3 protein stability.

23) Authors have claimed that METTL3 knockdown reduces the stability of FOXO3 protein. There are no mechanisms proposed in the field which link m6A or METTL3 to protein stability. Moreover, RNA stability is not directly correlated with the stability of a protein. Authors should suggest a possible explanation for the observation that METTL3 depletion affects FOXO3 protein stability.

24) In Figure 7I, authors should clarify the rationale behind the cutoffs chosen to define 'FOXO3 high' and 'FOXO3 low' cohorts. Why 'n' numbers are different in two groups?

25) Figures 5-7: The FOXO3 overexpression should have been performed with the control shRNA. Only one shMETTL3 is used. The PDXs should be used throughout and normal cells.

26) Please indicate the number of replicate in vivo studies.

27) I would encourage some in vivo studies with a survival endpoint. The differences in tumor sizes is somewhat modest. Do these changes generate survival differences?

Note: The grammar and writing warrant revision. There are too many claims of "significantly" without p-values. The "dramatically" and "remarkably" statements are overblown.

Statistical testing should be improved with correction for multiple comparisons. Please use SD, not SEM everywhere.

Referee #2:

In this work, the authors showed that downregulation of METTL3 was correlated with sorafenib resistance in hepatocellular carcinoma and a poor prognosis in clinic. METTL3 depletion promoted sorafenib resistance in hypoxic tumor microenvironment, which is accompanied by upregulated autophagy in liver cancer cells. This event is dependent on the FOXO3-METTL3 axis within hepatocellular carcinoma. Taken together, the authors provide evidence on the functional and mechanistic link involving m6A modification, sorafenib resistance and autophagy in hepatocellular carcinoma. I think the authors should address the following questions before this manuscript is suitable for publication.

Major Comments

1. In this work, transcriptome sequencing was performed in liver tumors and sorafenib-resistant ones (pages 4 and 5). Here the authors did not address the question whether sorafenib-resistant tumors were primary resistance or acquired resistance to sorafenib treatment. If primary resistance, liver tumors and sorafenib-resistant specimens were obtained from different individuals, how do the authors exclude the individual differences in sequencing data? If acquired resistance, whether downregulation of METTL3 is the main cause of acquired resistance and whether it is reasonable to associate METTL3 expression level with survival rates in HCC patients because METTL3 level is dynamic during the occurrence of sorafenib resistance?

2. The authors observed that 80% tumors exhibited high hypoxia level in HCC subcutaneous tumor model (page 5), and thus, indicated that oxygen deprivation was common in hepatocellular carcinoma. Was sorafenib resistance exclusively detected in hypoxic condition? In addition, the results showed that METTL3 depletion would increase the number of autophagosome under hypoxia (page 7, Fig 3A). Will there be any morphological changes in SMMC-7721 and METTL3 depletion cells under normal oxygen conditions? In addition, would hypoxia itself cause a significant increase in autophagy? A clear relationship between hypoxia condition, m6A modification and autophagy is lacking for the current manuscript.

3. In page 8 and Fig 4A, the authors showed that METTL3 KD would reduce the global m6A level of FOXO3, but did not show how m6A site was found in FOXO3 at single base resolution. Next, in Figs 4K and 4L, the authors found a potential m6A site in FOXO3 3'UTR, but there is no evidence to show that this site was the internal m6A site in FOXO3 and the authors did not provide experimental evidence that this site was m6A in vivo at single base detection level.

4. The authors mentioned that "YTHDF1-binding site was identified by ParCLIP at the 3'UTR of FOXO3", but the authors did not provide evidence that this site was identified by m6A reader YTHDF1 in vivo. The PAR-CLIP data cited might be a very different biological system. The authors performed YTHDF1 KD and found that FOXO3 was downregulated but these results did not prove that YTHDF1 is a reader of the m6A at the 3'UTR of FOXO3.

5. In Fig 4M, in calculating the relative luciferase activity, the control vector without any insertion should be used to normalize the translation differences between different cell lines. To confirm the effects of METTL3 and YTHDF1 knockdown, is there any rescue experiments after KD (e.g. rescue wild type METTL3 or catalytic dead METTL3)?

Minor comments

1. Some modifications should be made to the writing and grammar in this manuscript. In page 8, "YTHDF1-binding site" should be corrected with "YTHDF1-binding site". In page 11, "has not yet fully understood" should be corrected with "has not yet been fully understood". In page 11, "we were the first to demonstrated that" should be corrected with "we were the first to demonstrate that". In page 11, "may reverse the m6A-dependent" should be changed with "may reverse the m6A-dependent".

2. In Fig 1H, we noticed that n=23 in METTL3 Low group and that p value was lower than cutoff. Here I would like to ask whether the significant p value was due to too few statistical cases in METTL3 low group. The same question to Fig 7I.

3. In Fig 2E, please mark the molecular weight of METTL3 and GAPDH.

4. In Figs 2N and 2O, "Tubulin" should be right written in the figure legend.

5. In Figs 3B and 3D, the pattern of LC3-I and LC3-II was not consistent SMMC7721 and HepG2 cells (METTL3 KD group). There was an upregulation of both LC3-I and LC3-II in SMMC7721 after METTL3 KD, while LC3-I was downregulated and LC3-II was upregulated in HepG2 after METTL3 KD. Please offer an explanation.

6. In Fig 4B, "Aytophagy" should be corrected with "Autophagy" in the legend.

7. In Fig 4F, it would be much better to change the legend on X-axis into "Transcriptional Inhibition (h)", and change the legend on Y-axis into "Relative remaining RNA (%)". The maximum of Y-axis should be 100% but not 150%.

8. In Fig 5D, the right panel was an enlarge image of the corresponding left panel. Please mark the original position of the enlarging area in the left panel. The same question to similar imaging photos.

9. In Fig 5E, the images showed here were merged ones. Please provide some primary immunofluorescence images of the corresponding single channel.

10. In Fig 6E, in addition to tumor size data, is there any data to show the survival curves after implantation.

11. In Figs 6H to 6K, please offer an explanation of "Ki-67 relative expression (fold)" in IHC results. Usually, we do not use the relative expression level of specific markers in IHC slides. One possible approach was to categorize the results into low, medium, high, and very high levels according to staining intensity. Another feasible approach was to calculate the positive rates of cells in one standard area, and in this case, the maximum of Y-axis should no larger than 100%.

12. In Supplementary Fig 4C, please provide the western blot image in SMMC-7721 cells after FOXO3 overexpression.

13. In Supplementary Fig 5B, GAPDH, FOXO3 and METTL3 should be displayed in the same image.

Referee #3:

The manuscript by Lin et al examines the role of m6A methylation in liver cancer. They discovered that the catalytic subunit Mettl3 of the m6A holoenzyme is down-regulated in liver cancer that is resistant to treatment by sorafenib. They further make the case that hepatocellular carcinoma cells become resistant to sorafenib when Mettl3 is knocked-down and kept under hypoxia as in a tumor environment. Under these conditions, angiogenesis and autophagy are induced. They then identified Foxo3 as differentially regulated under these conditions and as target for m6A methylation. Lack of M6A mRNA methylation and concomitant reduction of FOXO3 decreased steady state transcript levels and decreased protein expression. They follow these observations up with overexpressing FOXO3 to show that it restored sensitivity to sorafenib. Most convincingly, they recapitulate then these cell culture findings in xenografts in a mouse model.

Overall, the paper is interesting and the data is solid and of high quality. The data support the observations and conclusions drawn. The paper can be published with minor revisions, outlined below, though it is not clear to me why the authors didn't chose a more cancer oriented journal.

Other:

P3. Why is liver cancer increasing?

English needs to be improved in many places, e.g.

"m6 A is a predominant internal modification of RNA

in mammalian cells with the feature of dynamic and reversibility"

"...the m6 A modification was in a lower level in sorafenib-resistant HCC..."

Background information on m6A, in particular the methylation complex and citations need to be updated e.g. a role for m6A in alternative splicing was first discovered in *Drosophila*. Many statements on m6A are imprecise, see below.

What is the core RNA demethylase?

YTH proteins, how do they specifically recognize m6A? "The m6 A status is governed by the m6 Abinding proteins such as YTH family member YTHDF1 and YTHDC1."

P6 line 11: Fig 2E should be 2D; Figure labelling needs to be accurate and checked!

P6. HUVEC?

Why they picked foxo3 and how it is linked to autophagy is not explained, and thus mysterious to non-experts.

Why was Foxo3 not up-regulated in Fig 1D?

Fig 5E I can't see a redistribution. This needs a higher magnification and quantification.

P10. This page is one paragraph. P6, p11 etc as well, this tool could be used more often to structure the text.

The figure legends need attention regarding English and accuracy of statements e.g.

Fig 4B legend: KEGG analysis shows that FOXO3 is involved in hypoxia and autophagy

Fig4C legend: Should state what was done rather than infer unproven conclusions.

I don't understand the model in Fig 7J. The model should summarize the findings such that it can be understood without text. Maybe splitting into the different scenarios will do the job.

1st Revision - authors' response

25 December 2019

Reviewer #1:

1. “The authors overuse cell lines, especially HepG2, which is often cross-contaminated. They use a patient-derived xenograft in a late experiment. As the claims are focused on hypoxia, it would be important to consider using more patient-derived materials and lines to avoid the effects of long term culture in room air.”

We thank Reviewer #1 for the suggestion on using more patient-derived materials and lines to support our findings. We extended the PDX experiments (Fig 7B-7H) and cell lines xenografts (Fig 6A-6C, S5G) in our revised manuscript to further validate the role of METTL3 in enhancing sorafenib-resistance in vivo. As for the cell lines used in this study, we previously authenticated the cell lines SMMC-7721, BEL-7402, HepG-2 and WRL68, and confirmed that they are not cross-contaminated in our laboratory.

2. “The exclusive focus on loss-of-function studies without METTL3 rescue or CRISPR studies limits confidence. It would be helpful to better develop controls. The in vivo studies use a single shRNA, which is an even lower threshold.”

We agree with Reviewer #1 that we need to include METTL3 rescue and CRISPR studies. In our revised manuscript, we generated shRNA-resistant METTL3 overexpression vectors and rescued the wild type form and catalytic mutant form of METTL3 in HCC cells with METTL3-depletion and sorafenib-resistant HepG-2 cells. We performed a series of functional experiments by rescue experiments, and showed that rescue of shRNA-resistant wild type METTL3 sensitized sorafenib-resistant HCCs to sorafenib treatment and reduced the accumulation of GFP-LC3 in the cells, as indicated in Fig 2E-2J, Fig S2M-S2Q, and Fig 3F-3H.

We also generated loss-of-function of METTL3 by CRISPR (Fig S5G) and examined the effects of METTL3-knockout in vivo (Fig 6A-6C), strengthening the results obtained by shRNA-mediated METTL3-knockdown in vivo.

3. “I would strongly suggest the use of over expression studies with METTL3 (and loss-of-function with FOXO3) in sorafenib-resistant HCC. The shRNAs for METTL3 have effects separate from sorafenib and METTL3 has been proposed as both an oncogene and tumor suppressor. Is METTL3 both necessary and sufficient?”

We thank Reviewer #1 for the suggestion. Overexpression of METTL3 in sorafenib-resistant HepG-2 cells rescued sorafenib-resistance and resensitized the cells to sorafenib treatment (Fig 2E, 2F, 2I and 2J), while overexpression of FOXO3 in sorafenib-resistant HepG-2 cells rescued the m6A-mediated sorafenib-sensitivity in HCC cells (Fig 5D, 5H, 5J and 5L). As knockdown of METTL3 in HCC showed increased resistance towards sorafenib treatment (Fig 2G, 2H, 2O and 2P), we suggest METTL3 plays both necessary and sufficient roles in mediating sorafenib-resistance in HCC.

4. “There was no comparison to normal liver cells. Is this a cancer-specific regulation? This would be important to consider as there needs to be consideration of toxicity.”

As suggested, we overexpress and knockdown METTL3 in normal liver cell line WRL68 (Fig S2A and S2B). Our results showed that the normal liver cells formed an enlarged colony after silencing METTL3 but had no meaningful change after overexpressing METTL3 (Fig 2A and 2B), indicating the effect caused by METTL3-depletion in part is not a cancer-specific regulation.

5. “The m6A studies are very interesting, but most of the biology is not really linked to m6A itself. It would be valuable to better develop the direct role of m6A in this process.”

We thank Reviewer #1 for the constructive suggestion. To determine the role of m6A in sorafenib-resistance in HCC, we rescued the wild type METTL3 and catalytic mutant METTL3 (resistant to METTL3 shRNA) in HCC cells with stable METTL3-depletion (Fig S2M and S2N) and sorafenib-resistant HepG-2 cells (Fig 2E and 2F). Our results showed that rescue of shRNA-resistant wild type METTL3 but not catalytic mutant METTL3 sensitized METTL3-knockdown HCC cells (Fig 2G, 2H, S2O and S2P) and sorafenib-resistant HepG-2 cells (Fig 2I, S2Q and 2J) to sorafenib treatment. In addition, rescue of wild type METTL3 but not catalytic mutant METTL3 reduced the accumulation of GFP-LC3 in these cells (Fig 3F, 3F-1, 3G, 3G-1, 3H, 3H-1, S3C and S3D), indicating the important and direct role of the m6A modification mediated by METTL3 in sorafenib-resistance in HCC.

6. “I was surprised that autophagy inhibitors were not used in validation.”

As suggested, the autophagy inhibitor 3MA was validated in HCC (Fig S3E).

7. “The authors use p-hacking for unbalanced groups in Figures 1H and 7I. This should be corrected.”

The survival analyses of the TCGA cohort involve two online tools, OncoLnc (<http://www.oncolnc.org/>) and R2 (<https://hgserver1.amc.nl/cgi-bin/r2/main.cgi>) a website of Genomics Analysis and Visualization Platform (It was mentioned in supporting materials and methods part). We did not use cBioportal for detailed analysis because cBioportal lacks customized setting and grouping patients with detailed clinical characterization. Specifically, Fig 1E was performed by OncoLnc with automatical default percentile I setting in LIHC, while Fig 1F (previous Fig 1H) and Fig 6M (previous Fig 7I) were performed by R2 in the condition of vascular invasion-micro involving the mechanism of sorafenib.

Reference:

1. Anaya J. 2016. OncoLnc: linking TCGA survival data to mRNAs, miRNAs, and lncRNAs. *PeerJ Computer Science* 2:e67 <https://doi.org/10.7717/peerj-cs.67>
2. R2: Genomics Analysis and Visualization Platform (<http://r2.amc.nl>)

8. “The data in Figure 1 is interesting but the clinical aspects should be better developed. The differences in patient response should be distinguished from other patient differences.”

As suggested, the data in Fig 1 has been reanalyzed to distinguish from other patient differences (Fig 1A-1C).

9. “The role of hypoxia is acceptable but somewhat premature. The claim is based on a single subcutaneous result without defining hypoxia or HIF function in the molecular regulation or cell biology. I was not sure why the experiment was not orthotopic, as well.”

As suggested, we have generated orthotopic liver tumor xenograft with Bel-7402 cells to validate the hypoxic condition in HCC. As shown in Fig S1F and S1F-1, higher level of HIF-1 α was detected in tumor tissue compared to non-tumor tissue, indicating oxygen deprivation was a common feature of HCC.

10. “All experiments presented in Figure 2 to 5 are performed under hypoxic conditions. These results establish that METTL3-FOXO3 axis regulates autophagy and sorafenib-resistance in HCC cell lines. Are these observations specific to hypoxia only? Authors are encouraged to perform key

mechanistic experiments under normoxic conditions as well.”

As suggested, we performed some key mechanistic experiments under normoxic conditions. We noticed the moderate increased IC₅₀ of METTL3-knockdown HCCs under normoxia condition (21% O₂) after treated with sorafenib (Fig S2J, S2K and S2L), excluding the potential molecular mechanism of METTL3-mediated sorafenib-resistance in HCC was mainly caused by hypoxia. Interestingly, we noticed the subcellular redistribution of GFP-LC3 by fluorescent immunostaining in normoxia condition and found a moderate increase of number in METTL3-depleted groups than their control groups (Fig S3A and S3B).

11. “Authors have used Crystal violet staining to measure cell viability. Since it is a qualitative assay, authors are suggested to perform an additional quantitative cell viability assay as well with appropriate statistics to support their findings.”

As suggested, we utilized CCK8 assay to quantitatively measure cell growth in HCCs with METTL3-knockdown or sorafenib-resistance. Our results showed that rescue of shRNA-resistant wildtype METTL3 but not catalytic mutant METTL3 sensitized METTL3-knockdown HCC cells (Fig S2O and Fig S2P) and sorafenib-resistant HepG-2 cells (Fig S2Q) to sorafenib treatment.

12. “On page 4, authors have stated “we performed transcriptome sequencing between liver tumors and sorafenib-resistant tumors to examine the differentially expressed genes”. It is not clear in the manuscript whether authors performed an RNA sequencing experiment? If yes then authors need to provide the details of number and type of samples used and also include the protocol in material and methods section. Additionally, it is important for the authors to elaborate the term “liver tumors and sorafenib-resistant tumors”.”

As noticed, to investigate the molecular mechanism of sorafenib-resistance in HCC, we obtained the acquired sorafenib-resistant liver tumors (n=3) from patients with long-term sorafenib-treatment, and performed transcriptome sequencing by comparing with sorafenib-sensitive liver tumors (n=3) to examine the differentially expressed genes. The RNA sequencing data have been uploaded in the database and the summary data have been attached in supplementary table 2. All the information has been revised in the manuscript.

13. “If RNA sequencing was not performed, then what is the source of transcriptome data shown in Figure 1? If authors have analyzed a previously available dataset then they should provide the reference as well as describe the method used to characterize sorafenib-resistant tumors from which data was obtained.”

We performed the RNA sequencing analysis from the acquired sorafenib-resistant liver tumors (n=3) from patients with long-term sorafenib-treatment and sorafenib-sensitive liver tumors (n=3). The RNA sequencing data have been uploaded in the database and the summary data have been attached in supplementary table 2. All the information has been revised in the manuscript.

14. “Please better define the “ctrl” for each shRNA experiment.”

All ctrls for each shRNA experiment or each overexpression experiment or each drug administration experiment have been changed to shCtrl, OE-Ctrl and DMSO, respectively. The manuscript has been revised accordingly.

15. “In supplementary Figure 1A, authors are suggested to use FDR in place of p-value to plot the data. It seems METTL3 lies on borderline of the cutoff and there are several other genes which are better candidates for follow up study. Authors are suggested to present a better explanation for the selection of METTL3 for the study.”

As suggested, Fig S1A has been revised. From the data analysis, we identified 820 genes with significant up-regulation and 776 genes with significant down-regulation in the sorafenib-resistant liver tumors from patients compared to the sorafenib-sensitive liver tumors from patients (Table S2). Gene enrichment analysis with KOBAS highlighted dysregulation of

several signaling pathways involved in drug-resistance, including cellular response to chemical stimulus, response to organic substance and response to nitrogen compound (Fig 1A). A novel group of transferase was identified to play roles in sorafenib-resistance in liver cancer. 13 genes were overlapped in these four signaling pathways (Fig 1B). We identified that METTL3, the primary component of m6A methyltransferase, was significantly down-regulated in sorafenib-resistant HCCs (Fig 1C), indicating that down-regulation of METTL3 may be implicated in sorafenib-resistance in HCC.

16. “In Figure 1B, authors are suggested to explain axis label and provide a meaningful axis title.”

As suggested, previous Fig 1B has been revised into Fig 1A accordingly.

17. “In Figure 1D, authors are suggested to provide values on color scale in place of 'min' and 'max'.”

As suggested, previous Fig 1D has been revised into Fig 1C accordingly.

18. “In Figure 1E, authors are suggested to depict the number of tumor samples analyzed in different groups.”

As suggested, previous Fig 1E has been revised into Fig 1D accordingly.

19. “In Figure 1F, it will be useful to include the data points on the Violin plots.”

As suggested, previous Fig 1F has been revised into Fig 1G accordingly.

20. “In Figure 1H, authors should clarify the rationale behind the cutoffs chosen to define 'METTL3 high' and 'METTL3 low' cohorts. Why 'n' numbers are different in two groups?”

The survival analyses of the TCGA cohort involve two online tools, OncoLnc (<http://www.oncolnc.org/>) and R2 (<https://hgserver1.amc.nl/cgi-bin/r2/main.cgi>) a website of Genomics Analysis and Visualization Platform (It was mentioned in supporting materials and methods part). We did not use cBioportal for detailed analysis because cBioportal lacks customized setting and grouping patients with detailed clinical characterization. Specifically, Fig 1F (previous Fig 1H) was performed by R2 with automatical default percentile setting in the condition of vascular invasion-micro involving the mechanism of sorafenib.

Reference:

1. Anaya J. 2016. OncoLnc: linking TCGA survival data to mRNAs, miRNAs, and lncRNAs. *PeerJ Computer Science* 2:e67 <https://doi.org/10.7717/peerj-cs.67>
2. R2: Genomics Analysis and Visualization Platform (<http://r2.amc.nl>)

21. “In supplementary Figure 1D, 1E and 1F, what is the meaning of "less hypoxia"?”

As noticed, “less hypoxia” has been changed into “normoxia”, where the expression levels of HIF-1 α were low in the tumor tissues.

22. “In Figure 2A, authors are suggested to provide a meaningful label for Y axis with proper units.””

As suggested, previous Fig 2A has been revised into Fig 1H accordingly.

23. “In figure 2, panel D and panel E have been interchanged. Authors are suggested to make the necessary corrections.””

As suggested, previous Fig 2D and 2E have been revised into Fig S2C and S2E accordingly.

24. “Figure 3: 3MA studies are interesting but lack proper controls of 3MA treatment with "ctrl", which I assume is a non-targeting control.””

As suggested, “ctrl” in 3MA studies in previous Fig 3 has been revised accordingly.

25. “On page 6, authors have stated that “Our results revealed that the tumor-conditioned medium (TCM) from HCC cells in the METTL3-knockdown groups showed decreased capacity to stimulate HUVECs tube formation compared to the control groups (Fig 2L and M)”. Actually, figure 2L and 2M show an opposite result. Authors are suggested to make the necessary corrections.”?”

As suggested, “decreased” has been revised into “increased” in the manuscript.

26. “In figure 4A, authors need to provide a meaningful label for the enrichment plots presented. It's unclear, what these enrichment plots are depicting? On page 7, authors describe these results as “Gene Set Enrichment Analysis (GSEA) was adopted to conduct the gene enrichment pathway in cellular response to stress, which was defined as unfavorable environmental conditions such as hypoxia”. Please elaborate.”?”

As noticed, the content has been revised into “Gene Set Enrichment Analysis (GSEA) was adopted to conduct the gene enrichment pathway in cellular response to stress, which was defined as unfavorable environmental conditions such as metabolism, osmotic stress and oxidative stress like hypoxia.”

27. “In figure 4C, authors have presented YTHDF1-ParCLIP data and m6A-seq data from SMMC-7721 cells and overlapped it with five other previously available m6A-seq datasets to demonstrate the presence of m6A peaks on FOXO3 mRNA. First, authors have not mentioned the source of YTHDF1-ParCLIP data used in this figure. Did authors perform YTHDF1-ParCLIP experiment? Second, it is not clear whether authors performed an m6A-seq experiment in SMMC-7721 cells. Authors have included a protocol for m6A sequencing in the materials and methods section of the manuscript but it describes that experiment were performed in Bel-7402 and HepG-2 cells but not in SMMC-7721 cells. Please clarify? Also, the m6A seq data obtained in Bel-7402 and HepG-2 cells are not used anywhere in the manuscript.”

As noticed, we performed the m6A-Seq to map the m6A site in SMMC-7721 cells under hypoxia, and the m6A sequence data has been uploaded in the database. We did not perform YTHDF1-ParCLIP sequencing, and the YTHDF1-ParCLIP data was obtained from the database GSE63591 (Wang et al. 2015) and the accession number has been added in Fig 4J (previous Fig 4C). However, we performed the m6A-RNA immunoprecipitation assay (MeRIP) and YTHDF1-RIP assay and analyzed with RT-PCR to validate FOXO3 as a bona fide target of METTL3 for the m6A modification under hypoxia (Fig 4M and 4N). The part of materials and methods has been revised accordingly.

Reference:

1. Wang X, Zhao BS, Roundtree IA, Lu Z, Han D, Ma H, Weng X, Chen K, Shi H, He C. N(6)-methyladenosine Modulates Messenger RNA Translation Efficiency. *Cell*. 2015, 161(6):1388-99
28. “In figure 4E, authors are suggested to present a quantification of immuno-bots and apply proper statistical methods to derive any conclusion about FOXO3 protein stability.”
- As suggested, the quantification of FOXO3 protein stability in Fig 4E (previous Fig 4E) has been shown in Fig 4E-1.
29. “Authors have claimed that METTL3 knockdown reduces the stability of FOXO3 protein. There are no mechanisms proposed in the field which link m6A or METTL3 to protein stability. Moreover, RNA stability is not directly correlated with the stability of a protein. Authors should suggest a possible explanation for the observation that METTL3 depletion affects FOXO3 protein stability.”

We thank Reviewer #1 for the constructive suggestion. After treated with cycloheximide, half-life of FOXO3 in protein level was remarkably decreased in METTL3-knockdown cells than in control cells, indicating that abolishment of the m6A methylation on FOXO3 could accelerate its degradation (Fig 4E and 4E-1). Interestingly, we detected a slightly increase of polysome processing in METTL3-knockdown cells under hypoxia (Fig S4D). By separating the RNA

fractions from non-translating fraction (<40S), translation initiation fraction (40S, 60S, 80S and <80S) to translation active polysomes (>80S), we found that knockdown of METTL3 significantly decreased translation efficiency of FOXO3 at 40S fraction (Fig 4F).

30. “In Figure 7I, authors should clarify the rationale behind the cutoffs chosen to define 'FOXO3 high' and 'FOXO3 low' cohorts. Why 'n' numbers are different in two groups?”

The survival analyses of the TCGA cohort involve two online tools, OncoLnc (<http://www.oncolnc.org/>) and R2 (<https://hgserver1.amc.nl/cgi-bin/r2/main.cgi>) a website of Genomics Analysis and Visualization Platform (It was mentioned in supporting materials and methods part). We did not use cBioportal for detailed analysis because cBioportal lacks customized setting and grouping patients with detailed clinical characterization. Specifically, Fig 6M (previous Fig 7I) was performed by R2 with automatical default percentile setting in the condition of tumor_status-with tumor.

Reference:

1. Anaya J. 2016. OncoLnc: linking TCGA survival data to mRNAs, miRNAs, and lncRNAs. *PeerJ Computer Science* 2:e67 <https://doi.org/10.7717/peerj-cs.67>
2. R2: Genomics Analysis and Visualization Platform (<http://r2.amc.nl>)

31. “Figures 5-7: The FOXO3 overexpression should have been performed with the control shRNA. Only one shMETTL3 is used. The PDXs should be used throughout and normal cells.”

As suggested, overexpression of FOXO3 in the control shRNA group was performed in Fig 5G, 5I, 5K and S5F accordingly. We utilized two sgRNAs to knockout METTL3 in xenograft mouse model, and showed consistent results with previous shRNA studies (Fig S5G and 6A-6C). We also extended our PDX models in vivo (Fig 7B-7H) and normal cell line WRL68 in vitro (Fig 2A-2B and S2A-S2B).

32. “Please indicate the number of replicate in vivo studies.”

As noticed, the information is listed in the figure legends.

33. “I would encourage some in vivo studies with a survival endpoint. The differences in tumor sizes is somewhat modest. Do these changes generate survival differences?”

We thank Reviewer #1 for the suggestion. As the tumor-bearing mice reach tumor volumes greater than 1500 mm³ will be killed according to regulation of the Animal Ethics Committee of Sun Yat-Sen University, we do not perform in vivo studies on the xenograft mouse model in our study. Instead, we extended our in vivo experiments by establishing a syngeneic xenograft mouse model in C57BL/6 mice with Hepa1-6 engineered with METTL3-knockout with CRISPR/Cas9 to further support our finding in this study (Fig 6A-6C).

34. “The grammar and writing warrant revision. There are too many claims of "significantly" without p-values. The "dramatically" and "remarkably" statements are overblown. Statistical testing should be improved with correction for multiple comparisons. Please use SD, not SEM everywhere.”

As noticed, the whole manuscript has been revised accordingly. SD was used in statistical testing.

Reviewer #2:

1. “In this work, transcriptome sequencing was performed in liver tumors and sorafenib-resistant ones (pages 4 and 5). Here the authors did not address the question whether sorafenib-resistant tumors were primary resistance or acquired resistance to sorafenib treatment. If primary resistance, liver tumors and sorafenib-resistant specimens were obtained from different individuals, how do the authors exclude the individual differences in sequencing data? If acquired

resistance, whether downregulation of METTL3 is the main cause of acquired resistance and whether it is reasonable to associate METTL3 expression level with survival rates in HCC patients because METTL3 level is dynamic during the occurrence of sorafenib resistance?"

We thank Reviewer #2 for the constructive question. As noticed, to investigate the molecular mechanism of sorafenib-resistance in HCC, we obtained the acquired sorafenib-resistant liver tumors (n=3) from patients with long-term sorafenib-treatment, and performed transcriptome sequencing by comparing with sorafenib-sensitive liver tumors (n=3) to examine the differentially expressed genes. The RNA sequencing data have been attached in supplementary table 2.

Interestingly, knockdown of METTL3 in HCC cells under normoxia condition and hypoxia condition showed increased IC50 towards sorafenib treatment (Fig 2 and Fig S2), excluding the potential molecular mechanism of METTL3-mediated sorafenib-resistance in HCC was mainly caused by hypoxia. Next, we utilized rescue experiments by overexpressing shRNA-resistant METTL3 in HCC cells with METTL3-depletion and sorafenib-resistant HepG-2 cells, and found that rescue of METTL3 resensitized the resistant cells to sorafenib treatment (Fig 3 and Fig S3), suggesting downregulation of METTL3 is the main cause of acquired resistance to sorafenib. By analyzing the pathological stage plot in liver cancer in TCGA database, we found that METTL3 expression level was significantly down-regulated in advanced stage of HCCs (Fig 1G). It indicates that down-regulation of METTL3 may be implicated in sorafenib-resistance in HCC.

2. "The authors observed that 80% tumors exhibited high hypoxia level in HCC subcutaneous tumor model (page 5), and thus, indicated that oxygen deprivation was common in hepatocellular carcinoma. Was sorafenib resistance exclusively detected in hypoxic condition? In addition, the results showed that METTL3 depletion would increase the number of autophagosome under hypoxia (page 7, Fig 3A). Will there be any morphological changes in SMMC-7721 and METTL3 depletion cells under normal oxygen conditions? In addition, would hypoxia itself cause a significant increase in autophagy? A clear relationship between hypoxia condition, m6A modification and autophagy is lacking for the current manuscript."

We thank Reviewer #2 for the constructive questions. Previous reports have shown that induction of HIF-1 α triggers a variety of cellular responses including autophagy under hypoxic condition (Azad et al, 2008; Rouschop et al, 2010; Tasdemir et al, 2008). In our revised manuscript, we performed some key mechanistic experiments under normoxic conditions. We noticed the moderate increased IC50 of METTL3-knockdown HCCs under normoxia condition (21% O₂) after treated with sorafenib (Fig S2J, S2K and S2L). Interestingly, we also noticed the subcellular redistribution of GFP-LC3 by fluorescent immunostaining in normoxia condition and found a moderate increase of number in METTL3-depleted groups than their control groups (Fig S3A and S3B). These results excluded the potential molecular mechanism by which METTL3 mediated sorafenib-resistance in HCC was mainly caused by hypoxia.

Reference:

1. Azad MB, Chen Y, Henson ES, Cizeau J, McMillan-Ward E, Israels SJ, Gibson SB. Hypoxia induces autophagic cell death in apoptosis-competent cells through a mechanism involving BNIP3. *Autophagy*. 2008, 4(2):195-204.
 2. Tasdemir E, Maiuri MC, Galluzzi L, Vitale I, Djavaheri-Mergny M, D'Amelio M, Criollo A, Morselli E, Zhu C, Harper F, Nannmark U, Samara C, Pinton P, Vicencio JM, Carnuccio R, Moll UM, Madeo F, Paterlini-Brechot P, Rizzuto R, Szabadkai G, Pierron G, Blomgren K, Tavernarakis N, Codogno P, Cecconi F, Kroemer G. Regulation of autophagy by cytoplasmic p53. *Nat Cell Biol*. 2008, 10(6):676-87.
 3. Rouschop KM, van den Beucken T, Dubois L, Niessen H, Bussink J, Savelkoul K, Keulers T, Mujcic H, Landuyt W, Voncken JW, Lambin P, van der Kogel AJ, Koritzinsky M, Wouters BG. The unfolded protein response protects human tumor cells during hypoxia through regulation of the autophagy genes MAP1LC3B and ATG5. *J Clin Invest*. 2010, 120(1):127-41.
3. "In page 8 and Fig 4A, the authors showed that METTL3 KD would reduce the global m6A level of FOXO3, but did not show how m6A site was found in FOXO3 at single base resolution. Next,

in Figs 4K and 4L, the authors found a potential m6A site in FOXO3 3'UTR, but there is no evidence to show that this site was the internal m6A site in FOXO3 and the authors did not provide experimental evidence that this site was m6A in vivo at single base detection level."

We thank Reviewer #2 for the question. We performed the m6A-Seq to map the m6A sites in SMMC-7721 cells under hypoxia, and found that the m6A site of FOXO3 locates at the 3'UTR region, which overlaps with an YTHDF1-binding site as previously identified by YTHDF1-RIP assay (Wang et al, 2015). Consistent with our results, previous studies identified FOXO3 mRNA as a direct target of the m6A modification (Batista et al, 2014; Lin et al, 2016; Meyer et al, 2012; Schwartz et al, 2014; Wang et al, 2014; Wang et al, 2015) as indicated in Fig 4J. Metagene analysis of FOXO3 at a spring-loaded base modification further confirmed the m6A site at the 3'UTR region (Fig S4E).

4. "The authors mentioned that "YTHDF1-binding site was identified by ParCLIP at the 3'UTR of FOXO3", but the authors did not provide evidence that this site was identified by m6A reader YTHDF1 in vivo. The PAR-CLIP data cited might be a very different biological system. The authors performed YTHDF1 KD and found that FOXO3 was downregulated but these results did not prove that YTHDF1 is a reader of the m6A at the 3'UTR of FOXO3."

We thank Reviewer #2 for the constructive suggestion. To prove the effect of m6A regulation on FOXO3 expression mediated by YTHDF1, we cloned the FOXO3 3'UTR into a dual luciferase reporter construct, and generated the mutant form of FOXO3 3'UTR by replacing A with T in the core motif (RRACH) of the potential m6A site (Fig 4K). Relative normalized luciferase activities of the wild-type and the mutant form of FOXO3 3'UTR were measured in Bel-7402 cells with various levels of METTL3. A remarkably reduction of luciferase activity was detected in the wild-type 3'UTR of FOXO3 in METTL3-knockdown and YTHDF1-knockdown cells. In contrast, for the mutant form of 3'UTR of FOXO3, depletion of METTL3 or YTHDF1 by its shRNA did not take any effects (Fig 4L). In METTL3-overexpressing cells, depletion of YTHDF1 abolished the induction of luciferase activity in the wild-type form of FOXO3 (Fig 4L). Further, to validate FOXO3 as a bona fide target of METTL3 for the m6A modification under hypoxia, we performed the m6A-RNA immunoprecipitation assay (MeRIP) and YTHDF1-RIP assay and analyzed with RT-PCR. Knockdown of METTL3 significantly reduced the m6A level of FOXO3 mRNA (Fig 4M), and reduced the binding activity of YTHDF1 with FOXO3 mRNA (Fig 4N). Our results suggested that METTL3 mediated m6A methylation at the 3'UTR of FOXO3 promoted FOXO3 mRNA stability in an YTHDF1-dependent manner.

5. "In Fig 4M, in calculating the relative luciferase activity, the control vector without any insertion should be used to normalize the translation differences between different cell lines. To confirm the effects of METTL3 and YTHDF1 knockdown, is there any rescue experiments after KD (e.g. rescue wild type METTL3 or catalytic dead METTL3)?"

As suggested, relative luciferase activity was normalized to the control vector without any insertion (Fig 4L). To confirm the effects of METTL3 knockdown, we rescued the wild type METTL3 and catalytic mutant METTL3 (resistant to METTL3 shRNA) in HCC cells with stable METTL3-depletion (Fig S2M and S2N) and sorafenib-resistant HepG-2 cells (Fig 2E and 2F). Our results showed that rescue of shRNA-resistant wild type METTL3 but not catalytic mutant METTL3 sensitized METTL3-knockdown HCC cells (Fig 2G and 2H) and sorafenib-resistant HepG-2 cells (Fig 2I) to sorafenib treatment. Consistently, the results of cell growth determined by CCK8 assay in HCCs with METTL3-knockdown (Fig S2O and Fig S2P) or sorafenib-resistance (Fig S2Q) showed similar rescuing tendency, as well as the rescue results in sorafenib-resistant HepG-2 cells by the cell survival assay (Fig 2J). In addition, rescue of wild type METTL3 but not catalytic mutant METTL3 reduced the accumulation of GFP-LC3 in these cells (Fig 3F, 3F-1, 3G, 3G-1, 3H, 3H-1, S3C and S3D). To confirm the effects of YTHDF1 knockdown, we rescued the wild type METTL3 (shRNA-resistant) in stable METTL3-depleted Bel-7402 cells. In METTL3-overexpressing cells, depletion of YTHDF1 abolished the induction of luciferase activity in the wild-type form of FOXO3 (Fig 4L). These results indicated the important role of the m6A modification mediated by METTL3 in sorafenib-resistance in HCC via an YTHDF1-dependent manner.

6. "Some modifications should be made to the writing and grammar in this manuscript. In page 8, "YTHDF1-binding site" should be corrected with "YTHDF1-binding site". In page 11, "has not yet fully understood" should be corrected with "has not yet been fully understood". In page 11, "we were the first to demonstrated that" should be corrected with "we were the first to demonstrate that". In page 11, "may reverse the m6A-dependent" should be changed with "may reverse the m6A-dependent"."

As noticed, the whole manuscript has been revised accordingly.

7. "In Fig 1H, we noticed that n=23 in METTL3 Low group and that p value was lower than cutoff. Here I would like to ask whether the significant p value was due to too few statistical cases in METTL3 low group. The same question to Fig 7I."

The survival analyses of the TCGA cohort involve two online tools, OncoLnc (<http://www.oncolnc.org/>) and R2 (<https://hgserver1.amc.nl/cgi-bin/r2/main.cgi>) a website of Genomics Analysis and Visualization Platform (It was mentioned in supporting materials and methods part). We did not use cBioportal for detailed analysis because cBioportal lacks customized setting and grouping patients with detailed clinical characterization. Specifically, Fig 1F (previous Fig 1H) and Fig 6M (previous Fig 7I) were performed by R2 with automatical default percentile setting in the condition of vascular invasion-micro involving the mechanism of sorafenib and the condition of tumor_status-with tumor, respectively.

Reference:

1. Anaya J. 2016. OncoLnc: linking TCGA survival data to mRNAs, miRNAs, and lncRNAs. *PeerJ Computer Science* 2:e67 <https://doi.org/10.7717/peerj-cs.67>
2. R2: Genomics Analysis and Visualization Platform (<http://r2.amc.nl>)

8. "In Fig 2E, please mark the molecular weight of METTL3 and GAPDH."

As noticed, Fig S2C (previous Fig 2E) has been revised accordingly.

9. "In Figs 2N and 2O, "Tubulin" should be right written in the figure legend."

As noticed, Fig 2N and 2O have been revised accordingly.

10. "In Figs 3B and 3D, the pattern of LC3-I and LC3-II was not consistent SMMC7721 and HepG2 cells (METTL3 KD group). There was an upregulation of both LC3-I and LC3-II in SMMC7721 after METTL3 KD, while LC3-I was downregulated and LC3-II was upregulated in HepG2 after METTL3 KD. Please offer an explanation."

The expression pattern of LC3-I and LC3-II varies in different cell types. An increase in LC3-II and a decrease in LC3-I expression is considered the biomarker of autophagy, as indicated in HepG-2 after METTL3 knockdown, but sometimes increases in LC3-II can be caused by enhanced autophagosome synthesis or reduced autophagosome recycling. The total amount of LC3-II could be evaluated and compared to a loading control as a means of assessing autophagy, as indicated in SMMC7721 cells with METTL3-depletion.

Reference:

1. Barth S1, Glick D, Macleod KF. Autophagy: assays and artifacts. *J Pathol.* 2010, 221(2):117-24.

11. "In Fig 4B, "Aytophagy" should be corrected with "Autophagy" in the legend."

As noticed, Fig 4B has been revised accordingly.

12. "In Fig 4F, it would be much better to change the legend on X-axis into "Transcriptional Inhibition (h)", and change the legend on Y-axis into "Relative remaining RNA (%)". The maximum of Y-axis should be 100% but not 150%."

As noticed, Fig 4D (previous Fig 4F) has been revised accordingly.

13. “In Fig 5D, the right panel was an enlarge image of the corresponding left panel. Please mark the original position of the enlarging area in the left panel. The same question to similar imaging photos.”

As noticed, Fig 5F (previous Fig 5D) and Fig 3A have been revised accordingly.

14. “In Fig 5E, the images showed here were merged ones. Please provide some primary immunofluorescence images of the corresponding single channel.”

As noticed, Fig 5G (previous Fig 5E), Fig 3F-3H, Fig S3A, Fig S3C, Fig S3D and Fig 5H have been revised accordingly.

15. “In Fig 6E, in addition to tumor size data, is there any data to show the survival curves after implantation.”

We thank Reviewer #1 for the suggestion. As the tumor-bearing mice reach tumor volumes greater than 1500 mm³ will be killed according to regulation of the Animal Ethics Committee of Sun Yat-Sen University, we do not perform in vivo studies on the xenograft mouse model in our study. Instead, we extended our in vivo experiments by establishing a syngeneic xenograft mouse model in C57BL/6 mice with Hepa1-6 engineered with METTL3-knockout with CRISPR/Cas9 to further support our finding in this study (Fig 6A-6C).

16. “In Figs 6H to 6K, please offer an explanation of “Ki-67 relative expression (fold)” in IHC results. Usually, we do not use the relative expression level of specific markers in IHC slides. One possible approach was to categorize the results into low, medium, high, and very high levels according to staining intensity. Another feasible approach was to calculate the positive rates of cells in one standard area, and in this case, the maximum of Y-axis should no larger than 100%.”

We utilized Image-Pro Plus software for pathology and IHC analysis. Ki-67 index of IHC slides from xenograft tumors was reassessed using a computer-based image analysis. In brief, of each specimen, five randomly chosen digital snapshots (high-power fields, 400×) were taken. Using Image-Pro Plus software, the number of pixels representing positively stained nuclei was detected, as well as the number of pixels representing the total nuclear area in one FOV. The index of proliferation was calculated by dividing the number of positive pixels representing the total nuclear area. The mean value of the five snapshots was used and expressed as the percentage of positive cells in each case. The results were compared and represented among different groups as previous reports (Samaroo et al. 2012; Wu et al. 2016).

Reference:

1. Samaroo HD, Opsahl AC, Schreiber J, O'Neill SM, Marconi M, Qian J, Carvajal-Gonzalez S, Tate B, Milici AJ, Bales KR, Stephenson DT. High throughput object-based image analysis of β -amyloid plaques in human and transgenic mouse brain. *J Neurosci Methods*. 2012, 204(1):179-188.
 2. Wu FQ, Fang T, Yu LX, Lv GS, Lv HW, Liang D, Li T, Wang CZ, Tan YX, Ding J, Chen Y, Tang L, Guo LN, Tang SH, Yang W, Wang HY. ADRB2 signaling promotes HCC progression and sorafenib resistance by inhibiting autophagic degradation of HIF1 α . *J Hepatol*. 2016, 65(2):314-24.
17. “In Supplementary Fig 4C, please provide the western blot image in SMMC-7721 cells after FOXO3 overexpression.”

As noticed, the western blot image in SMMC-7721 cells after FOXO3 overexpression was shown in Fig S5F.

18. “In Supplementary Fig 5B, GAPDH, FOXO3 and METTL3 should be displayed in the same image.”

As noticed, Fig S5H (previous Fig 5B) has been revised accordingly.

Reviewer #3:

1. “Overall, the paper is interesting and the data is solid and of high quality. The data support the observations and conclusions drawn. The paper can be published with minor revisions, outlined below, though it is not clear to me why the authors didn't chose a more cancer oriented journal.”

We thank Reviewer #3 for the positive comments, approbation of our work and recognition the novelty of the study.

2. “P3. Why is liver cancer increasing?”

The manuscript has been revised accordingly. “its incidence is increasing worldwide year by year compared with most solid tumors due to hepatitis viruses (HBV and HCV) infection and alcohol use (Siegel et al, 2019).”

Reference:

1. Siegel RL, Miller KD, Jemal A (2019) Cancer statistics, 2019. *CA: a cancer journal for clinicians* 69: 7-34
3. “English needs to be improved in many places, e.g. "m6 A is a predominant internal modification of RNA in mammalian cells with the feature of dynamic and reversibility" "...the m6 A modification was in a lower level in sorafenib-resistant HCC..."”

The manuscript has been revised accordingly.

4. “Background information on m6A, in particular the methylation complex and citations need to be updated e.g. a role for m6A in alternative splicing was first discovered in *Drosophila*. Many statements on m6A are imprecise, see below. 1). What is the core RNA demethylase? 2). YTH proteins, how do they specifically recognize m6A? "The m6 A status is governed by the m6 A binding proteins such as YTH family member YTHDF1 and YTHDC1."”

The “core” has been removed.

The m6A binding proteins with YT521-B homology (YTH) domain including YTHDC1, YTHDC2, YTHDF1, YTHDF2 and YTHDF3 recognize m6A in a methylation-dependent manner (Zaccara et al, 2019).

Reference:

1. Zaccara S, Ries RJ, Jaffrey SR (2019) Reading, writing and erasing mRNA methylation. *Nature reviews Molecular cell biology* 20: 608-624
5. “P6 line 11: Fig 2E should be 2D; Figure labelling needs to be accurate and checked!”

The manuscript has been revised accordingly.

6. “P6. HUVEC?”

The manuscript has been revised accordingly. HUVEC stands for human umbilical vein endothelial cell.

7. “Why they picked foxo3 and how it is linked to autophagy is not explained, and thus mysterious to non-experts.”

The members in FOXO family such as FOXO1, FOXO3, FOXO4 and FOXO6 (Jacobs et al, 2003) have been found to participate in various physiological response, such as DNA damage (Hien et al, 2002), caloric restriction (Carrano et al, 2009) and oxidative (Essers et al, 2005), and play crucial roles in cancer initiation, progression and chemo-resistance. Interestingly, previous reports have shown that FOXO3 upregulates autophagy in other tissues, including skeletal muscle (Mammucari et al, 2007) and bone (Gomez-Puerto et al, 2016). For example,

Mammucari et al. reported that FOXO3 controls the transcription of autophagy-marker genes, such as LC3 and BNIP3, and BNIP3 was the main player to mediate the effect of FOXO3 on autophagy in skeletal muscle (Mammucari et al, 2007). Gomez-Puerto et al. showed that FOXO3 induces autophagy in human mesenchymal stem cells in bone upon elevated ROS levels (Gomez-Puerto et al, 2016). Nonetheless, Schaffner et al. recently showed that FOXO3 controls autophagy in neuronal development, and FOXO3 deficiency induced autophagy to correct abnormal dendrite and spine development (Schaffner et al, 2018), which was consistent with our results that down-regulation of FOXO3 triggered autophagy signaling pathway in liver tumor tissue under intratumoral environment. These reports raise controversial debates on the role of FOXO3 in autophagy in various tissues and physiological conditions.

8. “Why was Foxo3 not up-regulated in Fig 1D?”

Gene enrichment analysis with KOBAS highlighted dysregulation of several signaling pathways involved in drug-resistance, including cellular response to chemical stimulus, response to organic substance and response to nitrogen compound (Fig 1A). A novel group of transferase was identified to play roles in sorafenib-resistance in liver cancer. 13 genes were overlapped in these four signaling pathways (Fig 1B). FOXO3 is not included in the overlapped group.

9. “Fig 5E I can't see a redistribution. This needs a higher magnification and quantification.”

Previous Fig 5E is replaced with Fig 5G.

10. “P10. This page is one paragraph. P6, p11 etc as well, this tool could be used more often to structure the text.”

The manuscript has been revised accordingly.

11. “The figure legends need attention regarding English and accuracy of statements e.g. Fig 4B legend: KEGG analysis shows that FOXO3 is involved in hypoxia and autophagy. Fig4C legend: Should state what was done rather than infer unproven conclusions.”

The figure legends have been revised accordingly.

12. “I don't understand the model in Fig 7J. The model should summarize the findings such that it can be understood without text. Maybe splitting into the different scenarios will do the job.”

A new working model figure has been revised accordingly (Fig 7J).

2nd Editorial Decision

24 February 2020

Thank you for submitting your revised manuscript for our consideration. It has now been seen once more by the three original referees (see comments below). As you will see, the referees acknowledge that the revised manuscript has improved, but they also find that a number of concerns that would need to be addressed remain.

We agree with the referees and appreciate that you have addressed many of the initial comments and provided additional experimental data. Thus we would like to give you the opportunity to address the remaining issues in an exceptional second round of revision. Here it will be important to 1) clarify if the sensitivity of normal liver cells to Sorafenib is also affected by METTL3 depletion or overexpression (Ref#1- point 1). 2) A panel showing FOXO3 levels for the METTL3 rescue experiments using the catalytic mutant (Ref#1- point 3 first part) should be added and 3) statistics for Figures 1F, 6G and 6M should be re-assessed and the comments of the referees (Ref#1- point 4, 5, Ref#2- point 4) incorporated into the discussion of these results. The remaining comments of referee #2 and #3 should be addressed by revising the text accordingly. As previously indicated, it is normally EMBO Journal's policy to allow only one round of experimental revision, such that it is now crucial that you fully resolve the three indicated main issues and carefully respond to or textually address the other concerns.

In addition to these specific points, I would also like to ask you to resolve several editorial issues, which are listed below in detail. If you have any questions regarding this revision or would like to discuss how to proceed in more detail, please feel free to contact me. I look forward to your revision.

REFeree REPORTS

Referee #1:

The revised manuscript by Lin et al presents an in depth interrogation of RNA m6A modification and m6A-methyltransferase METTL3 in sorafenib-resistance phenotype in hepatocellular carcinoma (HCC). Authors have established that METTL3 stabilizes FOXO3 mRNA in an m6A- and YTHDF1-dependent manner in Sorafenib sensitive HCC tumors and downregulation of METTL3 and FOXO3 confer resistance to Sorafenib by activation of autophagy in hypoxic tumor microenvironment. Authors have included several in vitro and in vivo experiments in the revised version of the manuscript to support the findings. Addition of wildtype and catalytic mutant METTL3 overexpression experiment, YTHDF1 RIP and polysome profiling has further validated the molecular mechanism of sorafenib resistance in HCC. The addition of in vivo study with PDX models further strengthen the therapeutic importance of the study. The manuscript provides a novel epitranscriptomic mechanism for Sorafenib resistance in HCC and hold potential therapeutic implications.

Major comments:

- 1) Authors should perform further experiments in METTL3 knockdown and overexpressing normal liver cells to demonstrate their response towards Sorafenib.
- 2) Authors are suggest to further explore the possible mechanism of increased colony size upon METTL3 knockdown in normal cells (Fig 2A). Does METTL3 plays a role in the growth of normal liver cells?
- 3) Authors should measure FOXO levels in wild type and catalytic mutant METTL3 rescue conditions in HCC cells with stable METTL3-depletion. Also, authors are suggested to perform YTHDF1 RIP in both wild type and catalytic mutant METTL3 rescue conditions to demonstrate that YTHDF1 binding to FOXO3 is impaired in catalytic mutant METTL3 overexpression. This will provide a direct evidence of the involvement of m6A in the regulation of FOXO3.
- 4) The issue of p-hacking still exists in the manuscript in Figure 1F and 6M. Authors have used OncoLnc to generate survival curves for these figures which provides an opportunity to manually set the percentile for low and high expression of a gene. The unbalanced groups arise due to the selection of different percentile values for low vs high expression. Authors are suggested to use bottom quartile vs top quartile for this analysis and generate unbiased survival curves with balanced groups. These panels must be removed.
- 5) Even in OncoLnc survival analysis of METTL3 in figure 1E, the p-value comes significant only when bottom third is compared with top third (33:33). Comparison of bottom quartile vs top quartile (25:25) yields a p value 0.0539, comparison of bottom half vs top half (50:50) yields a p value 0.219 whereas comparison of bottom 10% vs top 10% (50:50) yields a p value 0.141. Thus, it will be wise to either move this data to supplementary or remove from the manuscript.

Referee #2:

Thanks for the revised manuscript entitled "RNA m6A methylation regulates sorafenib-resistance in liver cancer through FOXO3-mediated autophagy". This work revealed a critical role of m6A modification within hepatocellular carcinoma under hypoxia conditions and the authors also made efforts in deciphering the molecular mechanism of RNA modification in sorafenib-resistance

hepatocellular carcinoma therapy. This revised version has satisfactorily answered the questions mentioned before while it is suitable for publication after the following questions are addressed.

1. English needs to be improved in some places. Please rewrite the last sentence of the introduction chapter. "Taken together, for the first time, our study revealed a crosstalk of the m6A modification, sorafenib-resistance and autophagy in hypoxia condition, providing insights into the multiple molecular mechanisms of sorafenib-resistance in HCC and expanding our understanding of therapeutic resistance".
2. It should be noticed that 6 in m6A is superscripted and this writing should be consistent within the manuscript.
3. In methods section, please confirm the right concentration of Polybrene.
4. In Fig 6G, it is not clear about the "statistical analysis results". It is difficult to figure out the correspondence between asterisks and the experimental groups.

Referee #3:

The authors have mostly addressed my comments and those of the other reviewers adequately. Before publishing, however, the m6A holoenzyme complex needs to be described in more detail in the introduction and the model in Fig 7 needs to be corrected, as Mett13 on its own is inactive.

2nd Revision - authors' response

13 March 2020

Reviewer #1:

7. "Authors should perform further experiments in METTL3 knockdown and overexpressing normal liver cells to demonstrate their response towards Sorafenib."

We thank Reviewer #1 for the suggestion on the role of METTL3 in response to sorafenib treatment in normal liver cells. Our new data showed that neither METTL3-knockdown nor METTL3-overexpression WRL68 cells affected sensitivity towards sorafenib treatment, indicating that sorafenib was not toxic to normal liver cells regardless of various levels of METTL3 (Appendix Fig S2C).

8. "Authors are suggest to further explore the possible mechanism of increased colony size upon METTL3 knockdown in normal cells (Fig 2A). Does METTL3 plays a role in the growth of normal liver cells?"

To explore the possible mechanism of increased colony size upon METTL3 knockdown in normal cells, we applied two assays (CKK8 assay and cell number counting assay) to examine the growth of normal liver cells with various levels of METTL3. Our new results demonstrated that knockdown of METTL3 in WRL68 cells moderately promoted cell growth while overexpression of METTL3 in WRL68 cells had no effects by cell proliferation assay and cell number counting (Appendix Fig S2D and E).

9. "Authors should measure FOXO levels in wild type and catalytic mutant METTL3 rescue conditions in HCC cells with stable METTL3-depletion. Also, authors are suggested to perform YTHDF1 RIP in both wild type and catalytic mutant METTL3 rescue conditions to demonstrate that YTHDF1 binding to FOXO3 is impaired in catalytic mutant METTL3 overexpression. This will provide a direct evidence of the involvement of m6A in the regulation of FOXO3."

We quite agree with Reviewer #1's suggestion. Our new results showed that rescue of wild-type METTL3 but not catalytic mutant METTL3 recovered the expression level of FOXO3 in METTL3-knockdown HCC cells (Appendix Fig S4D and E) and sorafenib-resistant HepG-2 cells (Appendix Fig S4F), suggesting the role of m6A-modification mediated by METTL3 in regulating FOXO3 expression level.

Meanwhile, rescue of wild-type METTL3 in METTL3-knockdown HCC cells significantly restored the binding amount of FOXO3 mRNA with YTHDF1 while rescue of mutant METTL3 only showed a slight increase binding amount compared to the control (Appendix Fig S4I), suggesting that METTL3 mediated m6A methylation at the 3'UTR of FOXO3 promoted FOXO3 mRNA stability in an YTHDF1-dependent manner.

10. “The issue of p-hacking still exists in the manuscript in Figure 1F and 6M. Authors have used OncoLnc to generate survival curves for these figures which provides an opportunity to manually set the percentile for low and high expression of a gene. The unbalanced groups arise due to the selection of different percentile values for low vs high expression. Authors are suggested to use bottom quartile vs top quartile for this analysis and generate unbiased survival curves with balanced groups. These panels must be removed.”

As suggested, previous Figure 1F and 6M have been removed from the manuscript.

11. “Even in OncoLnc survival analysis of METTL3 in figure 1E, the p-value comes significant only when bottom third is compared with top third (33:33). Comparison of bottom quartile vs top quartile (25:25) yields a p value 0.0539, comparison of bottom half vs top half (50:50) yields a p value 0.219 whereas comparison of bottom 10% vs top 10% (50:50) yields a p value 0.141. Thus, it will be wise to either move this data to supplementary or remove from the manuscript.”

We thank Reviewer #1 for the suggestion. Figure 1E has been moved to supplementary data (Appendix Fig S1B) and we also input the percentile of setting in the supplementary figure legend to show a clear comparison between these two groups.

Reviewer #2:

19. “English needs to be improved in some places. Please rewrite the last sentence of the introduction chapter. “Taken together, for the first time, our study revealed a crosstalk of the m6A modification, sorafenib-resistance and autophagy in hypoxia condition, providing insights into the multiple molecular mechanisms of sorafenib-resistance in HCC and expanding our understanding of therapeutic resistance”.”

As suggested, we have rephrased the sentence as “Taken together, our study first revealed a connection of the m6A modification, sorafenib-resistance and autophagy under hypoxia, and provided insights into the multiple molecular mechanisms of sorafenib-resistance in HCC as well as expanded our understanding of therapeutic resistance.”.

20. “It should be noticed that 6 in m6A is superscripted and this writing should be consistent within the manuscript.”

As suggested, all 6 in m6A in the manuscript have been revised to superscripted.

21. “In methods section, please confirm the right concentration of Polybrene.”

As noticed, “8 mg/mL” has been revised to “8 μ g/mL”.

22. “In Fig 6G, it is not clear about the “statistical analysis results”. It is difficult to figure out the correspondence between asterisks and the experimental groups.”

As suggested, Fig 6G has been revised into more clear format as well as Fig 6B.

Reviewer #3:

13. “The authors have mostly addressed my comments and those of the other reviewers adequately. Before publishing, however, the m6A holoenzyme complex needs to be described in more detail in the introduction and the model in Fig 7 needs to be corrected, as Mettl3 on its own is inactive.”

We thank Reviewer #3 for the suggestion. The content of the m6A holoenzyme complex has been added into the introduction part. “The m6A methyltransferase function of METTL3 strictly requires METTL14 as a co-activator. METTL3 and METTL14 form a m6A holoenzyme complex, where METTL3 functions as the catalytic subunit while METTL14 binds to RNA substrates, stabilizes the structure of the complex and activates METTL3 via allostery (Sledz & Jinek, 2016; Wang et al, 2016a; Wang et al, 2016b).”

And, the model in Fig 7J has been revised accordingly.

3rd Editorial Decision

24 March 2020

Thank you for submitting your revised manuscript and addressing the remaining referee concerns. I would now like to ask you to address a number of additional editorial issues that are listed in detail below in a final revised version. Please make any changes to the manuscript text in the attached document only using the "track changes" option. Once these remaining issues are resolved, we will be happy to formally accept the manuscript for publication.

3rd Revision - authors' response

25 March 2020

The authors performed all minor editorial changes.

Corresponding Author Name: Guohui Wan

Journal Submitted to: The EMBO Journal

Manuscript Number: EMBOJ-2019-103181R1